# Leveraging multinational enterprises to reduce the escalating regional carbon inequality in China

Kailan Tian [1,2,8], Yu Zhang [3,8], Jing Meng [4] ✉, Zhuoying Zhang [5] ✉, Yuli Shan [6], Heran Zheng [4], Xiaowei Nie[5,7] & Cuihong Yang [1,2] ✉

Multinational enterprises (MNEs) affect regional inequality as they exert substantial yet uneven economic and environmental effects. This study evaluates the impacts of MNEs on China's regional carbon emission inequality against value-added gains using an environmentally extended interprovincial input-output model that differentiates MNEs activities across China's provincial-level administrative divisions. We find that MNEs predominantly concentrated in developed coastal regions from 1997–2012, while less developed inland regions accounted for less than 30% of MNEs-generated value-added but over 50% of MNEs-related carbon emissions. This imbalance exacerbated regional inequality during this period. A pivotal change occurred over 2012–2017 as MNEs increasingly relocated inland and expanded their presence in clean technology-intensive industries, significantly reducing regional inequality. We also highlight the potential of leveraging MNEs' knowledge spillovers to mitigate emissions and inequality. These insights offer policy implications for strategically deploying MNEs to address inequality and climate change, aligning with Sustainable Development Goals.

Addressing the United Nations' Sustainable Development Goals (UN SDGs) 10 (reducing inequalities) and 13 (climate action) are critical and challenging[1,2]. Both goals call for intensified efforts from governments, enterprises and individuals[1,3–5]. In particular, multinational enterprises (MNEs) have been thrust to the forefront of both economic interaction and climate action[6,7] in the era of global production networks or global value chains (GVCs)[8–11] due to the profound impacts of MNEs transcending international borders. The contributions of MNEs to boosting economic development have been well recognized, and their carbon emission burdens have garnered increasing attention[6,7,12]. However, their effects on regional carbon inequality—the inequality in carbon emissions across different regions relative to their value-added gains—remain inadequately understood. Our research bridges this gap by providing a comprehensive analysis and offering insights into how MNEs can be strategically leveraged to simultaneously reduce regional inequalities and mitigate carbon emissions, thereby contributing to the achievement of the SDGs.

China has achieved remarkable success in attracting substantial MNEs since its opening-up policy and has emerged as the second largest host country of foreign direct investment (FDI) inflows[13]. MNEs may contribute to China's economic growth miracle through multiple channels, including the establishment of new production activities, the

[1]State Key Laboratory of Mathematical Sciences, Academy of Mathematics and Systems Science, Chinese Academy of Sciences, Beijing, China. [2]School of Economics and Management, University of Chinese Academy of Sciences, Beijing, China. [3]Yangtze Industrial Development Institute, Nanjing University, Nanjing, China. [4]The Bartlett School of Sustainable Construction, University College London, London, UK. [5]National Tibetan Plateau Data Center, State Key Laboratory of Tibetan Plateau Earth System, Environment and Resources (TPESER), Institute of Tibetan Plateau Research, Chinese Academy of Sciences, Beijing, China. [6]School of Geography, Earth and Environmental Sciences, University of Birmingham, Birmingham, UK. [7]School of Ecology and Environment, Tibet University, Lhasa, China. [8]These authors contributed equally: Kailan Tian, Yu Zhang. ✉e-mail: jing.j.meng@ucl.ac.uk; zhangzy@itpcas.ac.cn; chyang@iss.ac.cn

generation of multiplier effects via inter-industry linkages, technology transfers, and human capital accumulation[14]. However, MNEs and their effects were not evenly distributed across regions within China which is a vast country characterized by significant regional disparities. The economically developed Eastern China possesses a high capability to attract MNEs due to its prestigious geographical location, pioneering supporting policies, and favourable business environment[15]. MNEs situated in a specific region can further propagate their effects along China's domestic supply chains to an expanded range of regions[16]. Consequently, MNEs could potentially exert diverse economic and environmental impacts on different regions due to differences in economic development stages, local production structures, production technologies, etc. Quantifying these inequal effects across regions is crucial for comprehending the role of MNEs in China's regional carbon inequality.

Three primary strands of literature are relevant to our study. The first strand of literature focused on the inequality of economic benefits and environmental burdens embodied in international[17–21] or intranational trade[22–24]. These studies conducted in-depth analyses of the trade-climate dilemma[25–27] that developed economies[28] with more stringent environmental regulations and cleaner production technology transfer production activities with low value-added but high pollutant emission to developing economies[28] through trade, leading to emission leakages and economic and environmental inequality. The second strand extended the concentration from trade to FDI and has traced the carbon footprints of MNEs at both national[6,29,30] and global levels[7,12]. The third strand has estimated the inequality of carbon emissions across different income groups[31–36] or age groups[37–39].

In this work, we contribute to the literature by estimating how and to what extent MNEs in China contribute to its provincial value-added creation, carbon emissions, and the ensuing regional carbon inequality over the period of 1997–2017. We employ an interprovincial input-output (IPIO) model (Supplementary Table 1) and the latest IPIO tables[14] (Supplementary Note 1) that distinguish China's enterprises in each province-industry pair into three types by ownership, which are mainland China-, Hong Kong, Macao, and Taiwan- (HMT-), and foreign-owned enterprises, respectively. We also examine how MNEs may affect emission coefficient (emissions per unit of economic output) using econometric models and explore their potential to reduce the emission coefficient, emission volume, and carbon inequality. In pursuit of carbon neutrality and green development, China has released a circular outlining a plan to establish a dual control system for carbon emissions, with a primary focus on emission intensity (emissions per unit of value-added) supplemented by the control over total emission amount[40]. According to the circular, the nation will further improve its

emission calculation and carbon footprint accounting for regions, sectors and key enterprises. Responsibilities for dual control will be reasonably allocated to each region, key industries and enterprises. In this context, as one of the few studies delving into the effects of MNEs at the provincial level, our study contributes to the understanding of how MNEs may facilitate the achievement of dual control goals. Our findings suggest pertinent strategies for enhancing the effective use of FDI to achieve the goals of inequality reduction and emission mitigation, which both are pivotal components of the UN SDGs.

## Results
### Contribution of MNEs to value-added and carbon emissions
Investments from MNEs in China demonstrated a consistent upward trajectory from 1997 to 2017, significantly contributing to the country's explosive economic growth (Fig. 1a) and increased $CO_2$ emissions (Fig. 1b), particularly since China's entry into the World Trade Organization (WTO) in 2001. The value-added generated by MNEs soared from 1569 billion Yuan (the basic unit of the Chinese currency) in 1997 to a staggering 21,792 billion Yuan in 2017, with an impressive annual growth rate of up to 14.1%. Despite certain fluctuations over the 20 years, MNEs consistently contributed a significant proportion of China's national value-added, underscoring their enduring and substantial influence in China's economy. Specifically, MNEs contributed 19.7% of national value-added in 1997. This figure rapidly rose to 28.1% in 2002, the first year after China joined the WTO, and continued to increase, reaching 29.8% in 2007. It slightly dipped to 25.4% in 2012 before rebounding to 26.2% in 2017. From 1997 to 2017, $CO_2$ emissions generated by MNEs exhibited a surge of 265.6%, increasing from 744 million tonnes (Mt) in 1997 to 2721 Mt in 2017. Over this period, $CO_2$ emissions from MNEs accounted for approximately 20–30% of the total emissions from all enterprises of the three types of ownership, aligning with their contribution to value-added creation.

Our analysis reveals a significant shift in the sectoral contributions of MNEs from 1997 to 2017, paralleled by changes in their $CO_2$ emissions. With regards to value-added contribution at the sectoral level (Fig. 2a), the contribution shares of MNEs experienced an uptick in certain technology-intensive sectors like communication equipment (4.5% in 1997 and 7.6% in 2017) and transport equipment (3.1% in 1997 and 5.6% in 2017). Conversely, there was a slight decline in some traditional labour-intensive sectors such as textiles (10.0% in 1997 and 3.8% in 2017). These results are the comprehensive outcomes of the prevalence of MNEs within the sector, as well as the intersectoral linkages. With regards to $CO_2$ emission (Fig. 2b), MNEs generated the largest (around 50%) emissions in the production and supply of electricity, gas, and water. This highlights the role of upstream industries in

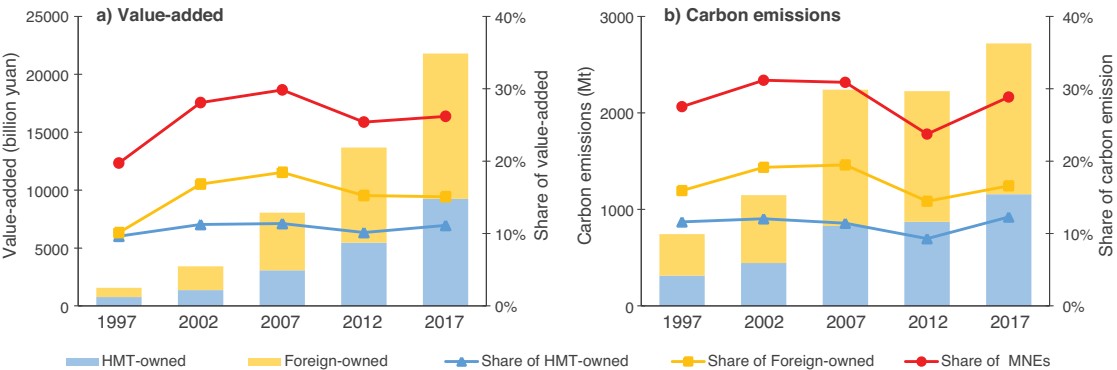

**Fig. 1 | Contribution of multinational enterprises to national value-added and CO₂ emissions. a** and **b** present the contribution of multinational enterprises (MNEs) to national value-added and $CO_2$ emissions, respectively. 'HMT-owned' and 'Foreign-owned' denote the contribution (absolute value) of Hong Kong, Macao, and Taiwan-owned (HMT-owned) and foreign-owned enterprises to national value-added or $CO_2$ emissions, respectively. 'Share of HMT-owned', 'Share of Foreign-owned', and 'Share of MNEs' give the shares of the contribution of HMT-owned, foreign-owned, and both types of MNEs in the national value-added or $CO_2$ emissions, respectively.

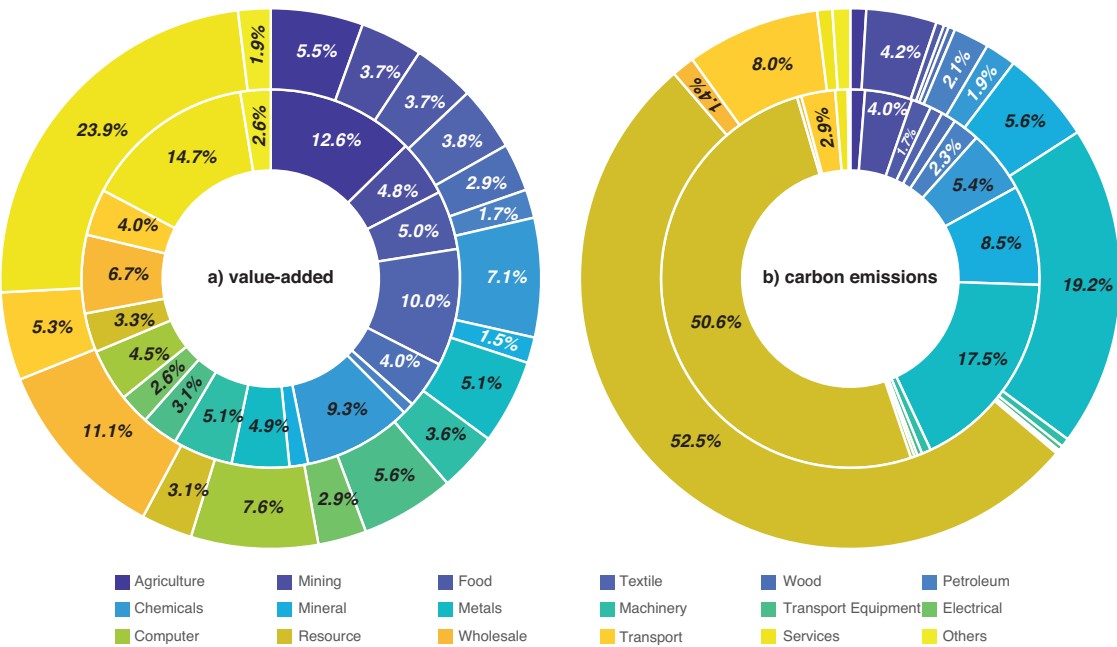

**Fig. 2 | Sectoral distribution of the contribution of multinational enterprises. a** and **b** present the sectoral distribution of the contribution of multinational enterprises (MNEs) to value-added and $CO_2$ emissions for 1997 (inner circle) and 2017 (outer circle), respectively. They provide the share of MNEs' contribution to a specific sector in MNEs' total contribution to all sectors. Sector descriptions are provided in Supplementary Table 2.

the overall carbon footprint of the economy. Other large emitters included the processing of metals and fabricated metal products, transport, storage and post, and mining and extraction of crude petroleum and natural gas. These sectors are all upstream and carbon-intensive industries (see Supplementary Table 2 for the detailed sector classifications).

MNEs generated substantial yet unequal (direct and indirect) value-added and $CO_2$ emissions across different provinces (China is officially divided into 34 province-level administrative divisions, the first level of administrative division in the country. For simplicity, these province-level administrative divisions are referred to as provinces[41]) over the 20 years (Fig. 3, see Supplementary Table 3 for the underlying data). Eastern China (see Supplementary Table 4 for the division of 31 provinces into four zones in mainland China) gained approximately 70% of China's total value-added generated by MNEs and was responsible for around 50% of the total national emissions generated by MNEs. Developed eastern provinces are characterized by advantageous geographical location, preferential policy support, robust business environment, and well-established industrial systems. These factors not only attracted MNEs directly but also fostered strong indirect contributions through interregional input-output linkages. In contrast, the other inland provinces (generally refer to central and western provinces that are geographically distant from the coast) held less than 30% (less than 1% for some remote inland provinces like Gansu, Qinghai, and Ningxia) of the total value-added generated by MNEs but were responsible for more than 50% of the total emissions generated by MNEs. Due to constraints such as high transportation costs, limited industrial foundation, and suboptimal policies, these provinces were not focal points for MNEs, leading to low direct contributions (often less than 5%). However, these provinces played a crucial role as essential suppliers of natural resources and upstream intermediate inputs which are more carbon-intensive. This enabled them to emit substantial indirect $CO_2$ for MNEs located in eastern provinces through interregional industrial linkages.

After a decade-long concentration of value-added generated by MNEs in Eastern China between 1997 and 2007, there was a notable transition in geographical distribution from the east to the west. This transition is evidenced by changes in the proportion of value-added generated by MNEs in Eastern China relative to the national total, which increased from 68.8% in 1997 to 72.0% in 2007, but subsequently declined to 67.6% in 2017. Between 2007 and 2017, the contribution of MNEs to local value-added increased in most central and western provinces, particularly in some southwestern regions such as Sichuan, Chongqing, and Guizhou. These changes were closely associated with the reallocation of FDI towards Central and Western China during the period. As industries matured, policies became more refined, and transportation infrastructure improved, a growing number of central and western provinces actively engaged in MNE-related activities.

## Carbon inequality relative to value-added gains

We propose the carbon Gini coefficient (C-Gini) as a metric for assessing the inequality in $CO_2$ emissions across China's regions relative to their value-added gains. Its scale ranges from 0 to 1. A value of 0 reflects the case of perfect equality, where each region within China contributes equally to $CO_2$ emissions per unit of value-added, while 1 denotes perfect inequality. We calculate two distinct C-Gini coefficients: the original C-Gini, which is derived from the original provincial value-added and $CO_2$ emissions, and the hypothetical one, which is computed using hypothetical values of provincial value-added and $CO_2$ emissions. The hypothetical values are obtained by extracting MNEs from the domestic economy (see the Methods for details). The original C-Gini reflects the current distribution of $CO_2$ emissions and value-added across provinces. The hypothetical C-Gini allows us to gauge how the absence of MNEs might influence this disparity.

China has undergone an escalating regional carbon inequality during 1997–2017, as evidenced by the rise in its original C-Gini coefficient from 0.22 in 1997 to 0.25 in 2007 and 0.32 in 2017 (Fig. 4). These C-Gini values underscore substantial discrepancies in emission intensity across China's provinces. Notably, some western provinces like Ningxia and Shanxi exhibited significantly higher emission intensities compared to some eastern provinces such as Guangdong and Fujian. In 2017, the emission intensities of Ningxia and Shanxi were 10 and 6 times higher than that of Guangdong, respectively. This disparity can

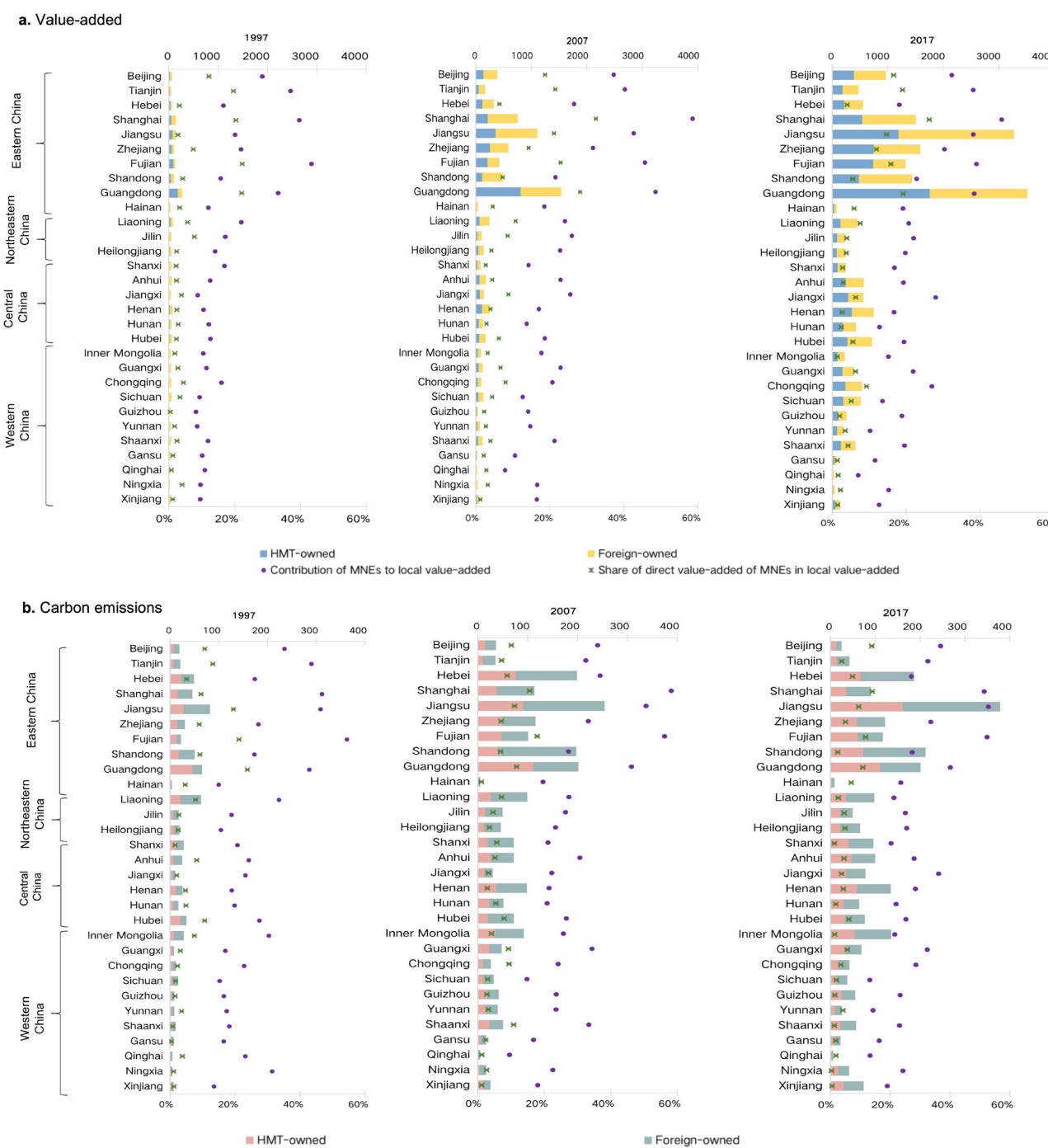

**Fig. 3 | Contribution of multinational enterprises to provincial value-added and CO$_2$ emissions. a** and **b** present the contribution of multinational enterprises (MNEs) to provincial value-added and CO$_2$ emissions for 1997, 2007, and 2017, respectively. The results for 2002 and 2012 are also available in Supplementary Table 3. 'HMT-owned' and 'Foreign-owned' denote the contribution of Hong Kong, Macao, and Taiwan-owned (HMT-owned) and foreign-owned enterprises to a province's value-added and CO$_2$ emissions, respectively. 'Contribution of MNEs to local value-added or carbon emissions' gives the proportion of the overall contribution of MNEs to a province's value-added or CO$_2$ emissions relative to the province's value-added or CO$_2$ emissions. 'Share of direct value-added or carbon emissions of MNEs in local value-added or carbon emissions' presents the share of direct value-added or CO$_2$ emissions of MNEs within a province in the province's value-added or CO$_2$ emissions. We do not present the results for Tibet because emission data for Tibet is not available.

be attributed to two key factors. On one hand, Western China has a higher proportion of carbon-intensive industries in its industrial structure due to its abundant natural resources such as coal, coke, oil, and gas. In contrast, Eastern China concentrates in cleaner manufacturing products and has a more service-oriented industrial structure. On the other hand, even within the same carbon-intensive industry, the emission intensity in Western China is significantly higher than that in Eastern China, reflecting the less advanced production technology in the west. We calculate the coefficient of variation (CV, the standard deviation divided by the mean) for each industry to quantify their emission intensity disparities across regions. Large CVs were observed in carbon-intensive industries, confirming that the emission intensity disparities in these sectors are crucial determinants of China's regional carbon inequality.

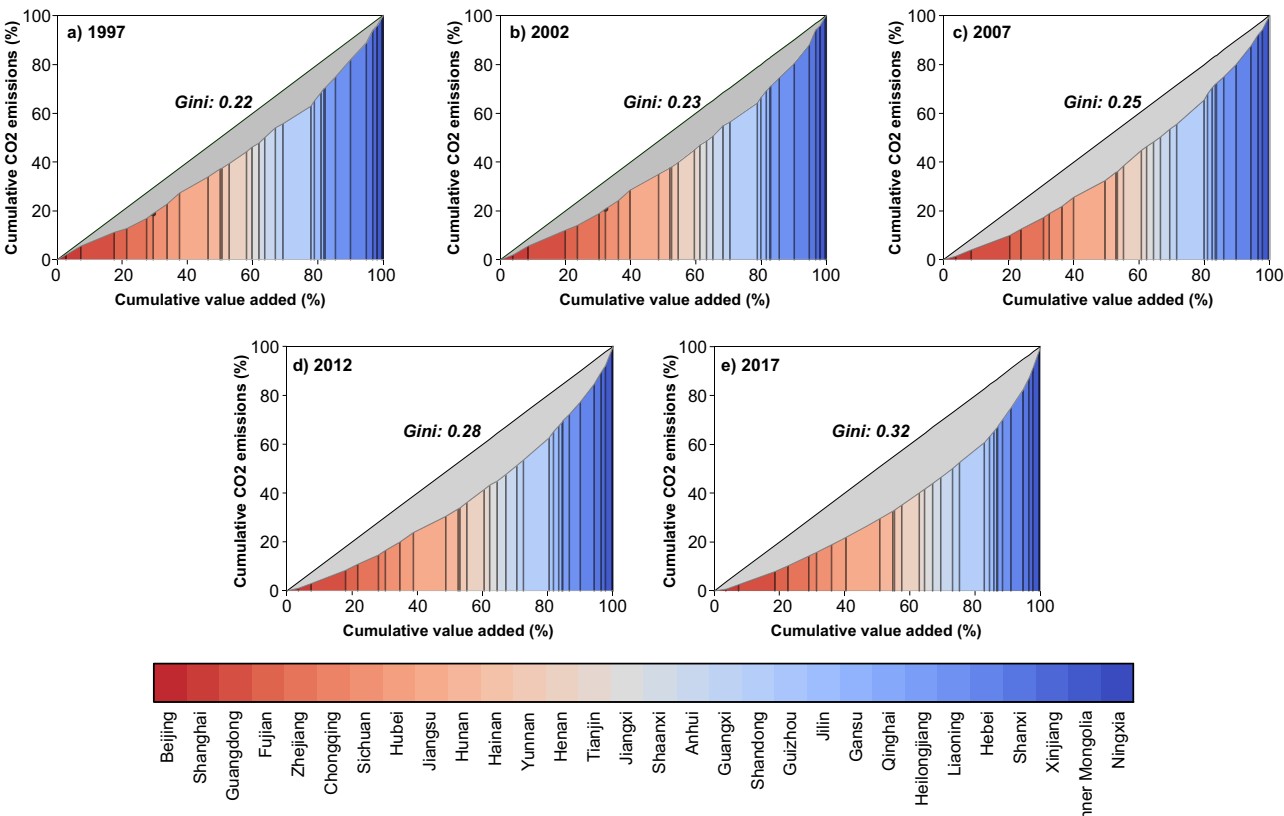

**Fig. 4 | China's carbon Gini coefficient changes from 1997 to 2017. a–d**, and **e** present China's original carbon Gini coefficients for 1997, 2002, 2007, 2012, and 2017, respectively. The provinces with different colours are ranked by emission intensity in 2017.

The exacerbated carbon inequality over the past 20 years can be attributed to varying economic development paces and widening gaps in emission intensity among regions. In 1997, the ten provinces (the rightmost ones in Fig. 4), which held 21.0% (accumulated 80–100%) of national value-added, emitted 35.0% of national $CO_2$ emissions. By 2017, although the emission share of these ten provinces had increased to 36.2%, their value-added share had declined to 17.8%. From 1997 to 2017, China experienced rapid economic development, leading to a substantial rise in the consumption of carbon-intensive products, such as energy and mineral products, particularly from the western regions. This dynamic resulted in a significant rise in $CO_2$ emissions in the West, where the growth rate was considerably higher than in the East. However, the ratio of economic growth to emissions growth in the West was lower than that in the East. For instance, from 1997 to 2017, Xinjiang's value-added increased by 928.6%, while its $CO_2$ emissions rose by 617.7%. In contrast, Jiangsu experienced a substantial 1,132.0% increase in value-added over the same period, accompanied by a comparatively moderate rise of 305.5% in $CO_2$ emissions. Consequently, the emission intensity gap between Xinjiang and Jiangsu widened from twofold in 1997 to fourfold in 2017.

From 1997 to 2012, the presence of MNEs was a driving factor that exacerbated China's carbon inequality. By 2017, a significant transformation occurred, with MNEs becoming a factor that mitigated this inequality. After extracting MNEs, the hypothetical C-Gini became 0.20 in 1997, 0.22 in 2002, 0.23 in 2007, 0.27 in 2012, and 0.34 in 2017. This indicates that the presence of MNEs amplified China's C-Gini during 1997–2012, but switched to decrease the C-Gini in 2017. Using a Structural Decomposition Analysis (SDA), we identify that the main driving factors behind this shift are changes in both the geographical distribution of MNEs and the structure of their products. In 1997, only 2.2% of MNEs located in Western China (6.1% in Central China and 7.0% in Northeastern China), contrasting sharply with the dominant presence of 84.7% of MNEs concentrated in Eastern China. Moreover, in Western China, a larger proportion of MNEs (32.6%) operated within carbon-intensive industries compared to domestically owned enterprises (23.5%). As a result, MNEs in Western China had higher overall emission intensities than domestically owned enterprises. In contrast, MNEs in Eastern China had significantly lower overall emission intensities compared to domestically owned enterprises. This widened the emission intensity gap between Western and Eastern China. This pattern persisted from 1997 to 2012, contributing to exacerbating carbon inequality during this period.

By 2017, the proportion of MNEs located in Western China had risen to 9.3%. Meanwhile, the share of MNEs in Western China operating in carbon-intensive industries had decreased to a lower level (16.2%). These changes led to the overall emission intensities of MNEs in Western China becoming significantly lower than those of domestically owned enterprises, thereby narrowing the emission intensity gap between Western and Eastern China. Consequently, the presence of MNEs became a factor that alleviated China's carbon inequality by 2017. The rise of MNEs in Central and Western China stems from two parallel trends: more MNEs establishing operations in these regions and coastal MNEs shifting their operations abroad. However, the first trend—new MNEs entry—had been the dominant force, as evidenced by the net increase in MNEs operating in China by 2017[14]. This indicates that the expansion of MNEs into inland areas, rather than offshoring from coastal regions, had played a more significant role in reshaping regional carbon inequality.

The significant changes in MNEs' geographical distribution and product structures observed since the 2010s are the combined consequences of both economic factors and policy interventions. First and foremost, production costs, which include labour, resources, and environmental costs, rose significantly in coastal regions since the 2010s, prompting MNEs to relocate to inland regions where

operational expenses remained lower. Second, China's increasingly stringent climate policies have incentivized cleaner production. For example, the '12th Five-Year Plan' set a target of a 17% reduction in $CO_2$ emissions per unit of Gross Domestic Product (GDP) from 2011 to 2015. Additionally, the 2012 'Plans for Greenhouse Gas Emissions Control in the 12th Five-Year Plan' proposed some specific mitigation strategies, including stricter regulations on energy-intensive industries by raising entry thresholds. These tightening regulations accelerated the adoption of emission-reducing technologies among MNEs and encouraged them to supply cleaner products. Third, China implemented regional development strategies, such as 'The Great Western Development Strategy' and 'The Rise of Central China Strategy', to reduce regional disparities[27]. These policies might also facilitate the relocation of MNEs to inland regions. It is noteworthy, however, that the ascendance of MNEs in the less developed regions has been relatively slow, exerting only a modest impact on regional inequality mitigation.

We assess the respective contributions of HMT-owned and foreign-owned enterprises to carbon inequality (Supplementary Table 5). The results indicate that foreign-owned enterprises had a slightly larger impact on the C-Gini, which aligns with the fact that there were relatively more foreign-owned enterprises than HMT-owned enterprises in China. Both types of MNEs influenced the C-Gini in the same direction, either increasing or decreasing it. This suggests that the distribution of both types of MNEs was similar across China's regions. Supplementary Table 5 also provides both the original and hypothetical C-Gini coefficients for the four zones in China. Notably, Western China had a high inequality level with the largest C-Gini (0.36 in 2017). This inequality was evident as some southwestern provinces, such as Sichuan, Chongqing, and Yunnan displayed significantly lower emission intensities compared to some northwestern provinces like Ningxia, Inner Mongolia, and Shanxi. Conversely, the C-Gini for Northeastern China hovered near zero, indicating an almost equal emission intensity level among the three northeastern provinces (Heilongjiang, Liaoning, and Jilin). With regards to the differences between hypothetical and original C-Ginis for the respective four zones, they present similar changes to China's pattern and can be explicated in a similar way.

We further examine which industries contribute most to the difference between the original and hypothetical inequality. We compare the original and hypothetical CVs for each industry and observe the largest differences between these two CVs in the following industries: general-purpose and special-purpose machinery (with a 9.0% change rate in 2007), transport equipment (7.2%), and communication equipment, computer, and other electronic equipment (6.7%). These industries possessed the highest proportion of MNEs, resulting in a larger change rate in the CVs after the extraction of MNEs. Nevertheless, it is important to note that these industries are relatively clean with lower emission intensity. Within these sectors, the emission intensity gaps between MNEs and domestically owned enterprises are not large. Consequently, compared to carbon-intensive industries, concentrating efforts on these sectors might offer a smaller potential for mitigating the overall regional inequality.

### Leveraging MNEs to reduce carbon inequality

We examine how MNEs in China may affect the emission coefficient. A decrease in the emission coefficient is widely regarded as an advancement in emission technology and is crucial for mitigating carbon emissions while sustaining economic growth[42]. To analyse this, we employ a panel regression model (see Supplementary Note 2 for a detailed introduction to our regression), using the extent of MNE presence in a province-industry as the core explanatory variable (MNE). This presence is measured by the ratio of the industry's intermediate consumption from MNEs to its overall intermediate consumption. We consider two distinct dependent variables: a province-industry's emission coefficient (ec) and the ratio of the emission

coefficient of domestically owned enterprises to that of MNEs within the same industry (gap). The second indicator measures the emission efficiency gap between domestically owned enterprises and MNEs. We address potential endogeneity and reverse causality by using lagged explanatory variables and an instrumental variable: province-level Foreign Market Access (FMA). This variable is measured by the distance from a province's capital city to its nearest port[43]. The rationale is that provinces closer to a port are more likely to have better access to foreign markets, making them more appealing to MNEs. Furthermore, a region's emission efficiency is not related to its geographical characteristics, making this an appropriate instrumental variable. Given that the main channels through which a region's emission performance is determined are scale effects, sectoral composition effects, and technique effects[44], we further introduce some control variables, including production scale, industrial structure, technology innovation level, urbanization level, and environmental regulation level[45].

Table 1 demonstrates that the presence of MNEs significantly reduces the emission coefficient and effectively narrows the emission coefficient gaps between domestically owned enterprises and MNEs. We validate the robustness of these results through several approaches: comparing outcomes without and with control variables (columns 1 and 2), employing robust standard errors clustered at the regional (column 3) and industry (column 4) levels, utilizing instrumental variables (column 5), and applying an alternative estimation method–the Generalized Method of Moments (GMM, column 6). These findings suggest that the presence of technologically advanced MNEs in China significantly bolsters the emission technology upgrading of local enterprises. Most MNEs in mainland China are owned by technologically advanced countries/regions such as the United States, European countries, Japan, and the Chinese regions of Hong Kong, Macao, and Taiwan. These MNEs significantly enhance China's access to advanced technologies and cleaner intermediate inputs from global markets. Moreover, they act as key conduits for transferring environmentally friendly technologies and advanced machinery from their home countries to China. Local enterprises observe, imitate, and adopt the innovative practices, technologies, and management techniques employed by MNEs. This process drives the upgrading of local enterprises' production capabilities and technological advancements, fostering greater efficiency and sustainability.

Next, we design scenarios to investigate the potential for reducing China's carbon emissions and mitigating the carbon inequality by leveraging the spillover effects of MNEs, with the aim of providing insights for relevant policymaking. Given the fact that the gaps in emission coefficients between domestically owned enterprises and MNEs are larger in inland regions than in Eastern China, we believe it is more feasible and urgent for domestically owned enterprises in inland regions to narrow their emission technology gaps with their multinational counterparts. Therefore, in our scenario settings, we prioritize upgrading emission technologies in inland regions. In the first scenario, we facilitate a convergence by adjusting the emission coefficients of domestically owned enterprises in carbon-intensive industries in Western China to align with those of MNEs within the same industry and province. If there are no MNEs present in the carbon-intensive industries of a particular province, adjustments are made by aligning the emission coefficients of domestically owned enterprises in that province with the average levels observed in MNEs across western provinces. The rationale behind this scenario lies in addressing the prominent cause of China's regional carbon inequality, namely, the higher emission intensity in Western China compared to Eastern China. By enabling Western domestic-owned enterprises to fully learn from local MNEs and upgrade their emission technology to meet identical standards, this scenario captures the potential for reducing inequality, with a specific focus on carbon-intensive industries in Western regions. When running the scenario based on the 2017 situation, the simulation results show a considerable reduction in carbon emissions by 832 Mt

**Table 1 | Multinational enterprises and emission coefficients**

| | (1) ec | (2) ec | (3) ec | (4) ec | (5.1) MNE | (5.2) ec | (6) ec |
|---|---|---|---|---|---|---|---|
| MNE | −0.012** | −0.017*** | −0.017*** | −0.017** | | −0.023*** | −0.017** |
| | (0.005) | (0.006) | (0.006) | (0.007) | | (0.010) | (0.008) |
| constant | 0.010*** | 0.029*** | 0.029*** | 0.029*** | 0.036*** | 0.015*** | 0.001 |
| | (0.001) | (0.005) | (0.005) | (0.009) | (0.011) | (0.004) | (0.006) |
| FMA | | | | | −0.007*** | | |
| | | | | | (0.001) | | |
| MNE (−5) | | | | | 0.312*** | | |
| | | | | | (0.033) | | |
| ec (−5) | | | | | | | 0.481*** |
| | | | | | | | (0.100) |
| Controls | No | Yes | Yes | Yes | Yes | Yes | Yes |
| Fixed Effect | Yes | Yes | Yes | Yes | Yes | Yes | Yes |
| AR (1) | | | | | | | [0.004] |
| AR (2) | | | | | | | [0.105] |
| Overidentification test | | | | | | | [0.901] |
| Observations | 3600 | 3023 | 3023 | 3023 | 2615 | 2615 | 2615 |
| $R^2$ | 0.263 | 0.316 | 0.316 | 0.316 | 0.728 | 0.376 | |
| | gap | gap | gap | gap | MNE | gap | gap |
| MNE | −2.409*** | −1.818*** | −1.818*** | −1.818*** | | −7.893** | −7.280*** |
| | (0.860) | (1.065) | (1.267) | (1.292) | | (3.797) | (2.181) |
| constant | 2.146*** | 0.5279** | 0.5279* | 0.5279* | 0.036*** | 0.477** | −0.613 |
| | (0.097) | (0.639) | (0.768) | (0.817) | (0.011) | (0.889) | (2.023) |
| FMA | | | | | −0.009*** | | |
| | | | | | (0.001) | | |
| MNE (−5) | | | | | 0.312*** | | |
| | | | | | (0.030) | | |
| gap (−5) | | | | | | | 0.390*** |
| | | | | | | | (0.112) |
| Controls | No | Yes | Yes | Yes | Yes | Yes | Yes |
| Fixed Effect | Yes | Yes | Yes | Yes | Yes | Yes | Yes |
| AR (1) | | | | | | | [0.000] |
| AR (2) | | | | | | | [0.108] |
| Overidentification test | | | | | | | [0.388] |
| Observations | 3600 | 3023 | 3023 | 3023 | 2615 | 2615 | 2615 |
| $R^2$ | 0.298 | 0.475 | 0.475 | 0.475 | 0.726 | 0.468 | |

Table 1 presents the industry-level regression results of the explanatory variables on emission coefficient (ec) and the emission coefficient gap (gap) between domestic-owned enterprises and multinational enterprises (MNEs), respectively. ***$p < 0.01$, **$p < 0.05$, *$p < 0.1$. Both the explanatory variables and dependent variables are in natural logarithms. The notation (−5) denotes a lag of 5 years. The sample covers 30 sectors for 30 provincial units (Tibet excluded due to data unavailability) in China over 5 years (1997, 2002, 2007, 2012, and 2017). When running the regressions, some observations with null values are excluded automatically. Columns 1 and 2 present the results without and with control variables, respectively, using robust standard errors in parentheses. Columns 3 and 4 present the results with robust standard errors clustered at the regional and industry levels, respectively. Columns 1–4 utilize the Least Squares estimation. Columns 5.1 and 5.2 present the results estimated using instrumental variables and the Two-Stage Least Squares (2SLS) estimation. Column 6 presents the results using the Generalized Method of Moments (GMM) estimation. We include province, industry, and time fixed effects in all the regressions. We consider five control variables, of which the regression results are provided in Supplementary Table 6.

and a decrease of 12.5% in China's C-Gini, declining from the original level of 0.32 in 2017 to 0.28 (Fig. 5a). These results underscore the effectiveness of this convergence strategy in achieving both environmental improvements and inequality reduction.

In the second scenario, we extend the scope to cover all industries and adjust the emission coefficients of domestically owned enterprises across all industries in Western China following the same approach as in the first scenario. We observe a decline in carbon emissions by 1250 Mt and a larger reduction of 18.8% in the C-Gini, decreasing from 0.32 to 0.26 (Fig. 5b). A comparison of the outcomes of the first and second scenarios highlights the importance of prioritizing the enhancement of emission mitigation technology in carbon-intensive industries. The reduction in the emission coefficients of these industries

demonstrates a higher marginal contribution towards mitigating both carbon emissions and carbon inequality.

In the third scenario, our focus extends beyond western provinces to encompass all provincial units in Western China, Central China, and Northeastern China, where emission technologies currently lag behind those in Eastern China. We simulate a situation where domestically owned enterprises in carbon-intensive industries in these regions have the opportunity to transition to the same emission technology as local MNEs. The simulation results demonstrate a substantial decline in carbon emissions by 2610 Mt and an impressive reduction of 31.3% in the C-Gini, decreasing from 0.32 to 0.22 (Fig. 5c).

In the fourth scenario, we consider the possibility that MNEs may not increase their investments in inland regions due to economic and geopolitical issues. In such a case, the expected technology spillover

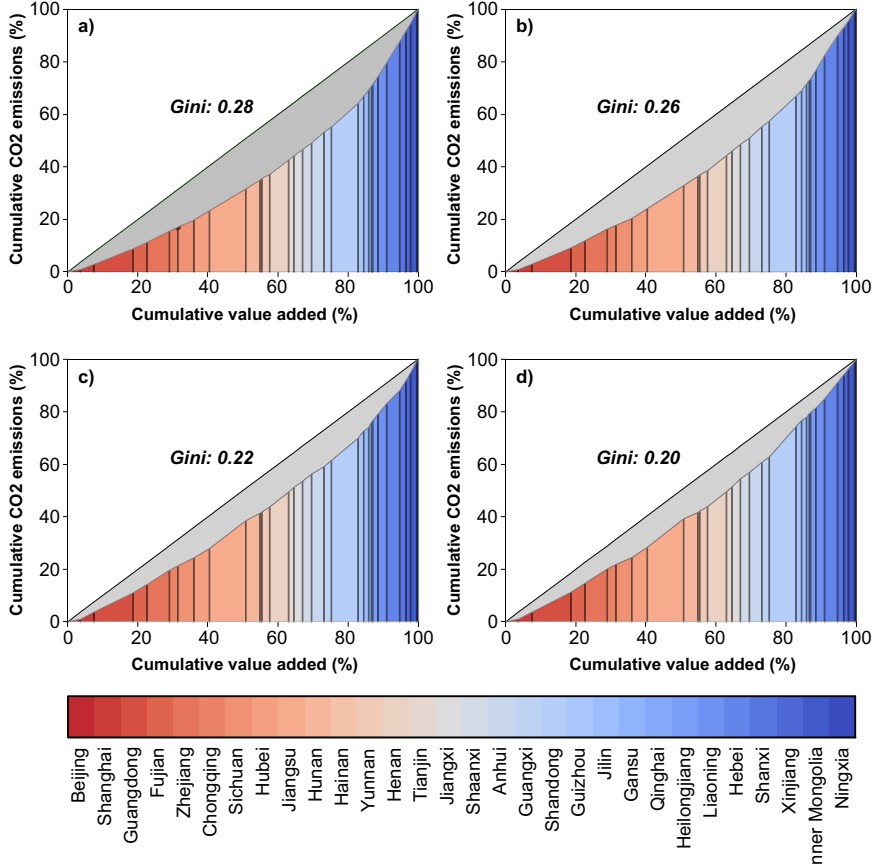

**Fig. 5 | China's carbon Gini coefficients in four scenarios. a** China's carbon Gini coefficients in the scenario where the emission coefficients of domestically owned enterprises in carbon-intensive industries in Western China converge to those of multinational enterprises (MNEs) within the same industry and province. **b** the scenario where the emission coefficients of domestically owned enterprises across all industries in Western China converge to those of MNEs. **c** the scenario where the emission coefficients of domestically owned enterprises in carbon-intensive industries in inland regions converge to those of MNEs. **d** the scenario where the emission coefficients of domestic-owned enterprises across all industries in inland regions converge to those of domestic-owned enterprises in Eastern China.

effects from MNEs on local enterprises in the previous three scenarios could be interrupted. As an alternative strategy, China could stimulate domestically owned enterprises in coastal regions to expedite the transfer of advanced technologies to inland regions. Therefore, we simulate a situation where domestically owned enterprises in inland regions converge to the emission technology levels of those in Eastern China. The results show that the carbon emissions would decrease by 2790 Mt and the C-Gini would decrease from 0.32 to 0.20 (Fig. 5d). These scenarios provide us with valuable insights into strategies for improving economic benefits and reducing carbon inequality, which will be discussed in detail in the following section.

## Discussion

Attracting MNEs has been an important pillar of economic development policy as most countries in the world have recognized their contributions to economic growth[46]. However, the effects of MNEs on carbon emissions and the ensuing carbon inequality are far from being adequately acknowledged. In this paper, we thus enrich the relevant research by conducting a thorough analysis of China. Over the past decades, China has successfully attracted substantial MNEs due to the acceleration of globalization and some inherent advantages, including its cost-effective labour, open trade and investment policies, and expansive consumer market. Our analyses indicated that MNEs generated substantial yet uneven value-added and emissions across different regions in China. We emphasize that MNEs represent a non-negligible yet frequently overlooked factor in shaping carbon inequality. They exacerbated China's escalating carbon inequality

between 1997 and 2012 but subsequently reduced it by 2017. Furthermore, our scenario analyses underscored the potential for China to significantly mitigate emissions and alleviate regional inequality by leveraging the positive spillover effects from MNEs.

However, China's attracting more MNEs have encountered new challenges. The global investment landscape has been seriously clouded by formidable challenges, including the slowing global economy in the aftermath of COVID-19, increasingly strained geopolitical relationships, and growing discourse of self-sufficiency[47]. Rising labour costs, geopolitical uncertainties, and concerns about the health of China's economy may also prompt MNEs to shift investment and their headquarters out of China. In light of these circumstances, it is imperative for China to make efforts not only to prevent the excessive exodus of MNEs but also to attract a growing influx of foreign investment. A practical strategy would involve fostering mutual benefit through strengthening political and economic ties with the United States, Europe, and other countries. Cultivating positive diplomatic relationships contributes to an investor-friendly global climate. Furthermore, policies that foster a transparent business environment, fair competition in government procurement, and robust intellectual property protection are essential for attracting and retaining MNEs within the country.

We emphasize the need for intensified efforts to attract more foreign investment to inland China. One important finding from our study is that the increasing presence of MNEs in inland regions has contributed to balanced economic development and mitigated regional carbon inequality. We also found that the ascendance of MNEs

in inland regions during 1997–2017 was relatively slow, resulting in a modest effect. Accelerating this trend could further amplify its positive effects on narrowing the gap between coastal and inland provinces and reducing inequality. As labour, resources, and environmental costs continue to rise in coastal provinces, reorienting investment towards inland regions emerges as a favourable strategy for MNEs. Nevertheless, facilitating this transition necessitates the implementation of targeted policies. Inland provinces are strongly suggested to improve their governance practices to cultivate a favourable business environment and augment incentives such as fiscal and taxation support to attract foreign investors.

Our research suggests that MNEs should share the carbon mitigation responsibility in their host regions as they generate a significant proportion of $CO_2$ emissions. We emphasize that establishing a sound carbon emission accounting system is a prerequisite for assigning the mitigation responsibilities to different regions, industries, and key enterprises, thereby supporting China's dual control efforts. Our scenario analyses further underscore the imperative to enhance low-carbon technologies in inland regions, particularly within carbon-intensive industries. Firstly, inland provinces should proactively strive to narrow their technology gap with coastal regions and the gap between domestically owned enterprises and MNEs. This entails boosting domestic investments and attracting more foreign capital for the development of low-carbon technologies. Strengthening coordination between MNEs and local enterprises is imperative to expedite the diffusion of cleaner production technology. Secondly, if MNEs reduce their investments in China or its inland regions, disrupting the technology spillover effects to local enterprises, it becomes crucial to stimulate domestically owned enterprises in coastal regions to increase their investments in clean technologies in inland areas. This will help accelerate the transfer of knowledge and advanced emission-reducing technologies to these regions[27]. Thirdly, a regional collaborative strategy is essential for China to reduce the escalating carbon inequality, given that a significant proportion of $CO_2$ from inland provinces is emitted for the production activities in coastal regions. The establishment of an expanded carbon market (including only 2162 coal-power plants[24] by 2023) that encompasses a broader range of carbon-intensive industries could serve as an effective mechanism to alleviate regional carbon inequality. This provides a significant opportunity for inland provinces, where many carbon-intensive industries are concentrated, to gain economic benefits with minimal additional emissions, thereby reducing carbon inequality.

This study focused on China and uncovered the relationships between its regional inequality and MNEs. Some findings and implications might be context-specific due to China's unique economic and institutional landscape. Future research may extend our methodology and analytical framework to conduct analysis for other countries. For countries with pronounced regional disparities, our approach may serve as a preliminary framework for examining subnational inequality. However, policy recommendations should be carefully tailored to local conditions, as socioeconomic, institutional, and technological factors vary widely across contexts. From a global perspective, promoting low-carbon technology transfer from MNEs to technologically disadvantaged countries could help mitigate carbon leakages and support emission reductions. Nevertheless, such efforts face practical challenges, including intellectual property barriers, divergent regulatory standards, and uneven technological absorption capacity. While these policy directions hold potential, their practical implementation demands careful contextual assessment and tailored strategies.

## Methods
### Tracing the contribution of MNEs
We adopt an interprovincial input-output (IPIO) model that distinguishes three types of enterprises by ownership in each province-industry pair[14],

which are mainland China-, HMT-, and foreign-owned enterprises, respectively (Supplementary Table 1). Mainland China is geographically disaggregated into 31 provincial units with $n$ industries in each province and three types of enterprises in each province-industry pair.

The market-clearing condition yields $x = Ax + Yu + p$, and the solution is given by Eq. (1):

$$\mathbf{x} = (\mathbf{I} - \mathbf{A})^{-1}(\mathbf{Yu} + \mathbf{p}) = (\mathbf{I} - \mathbf{A})^{-1}\mathbf{f} \tag{1}$$

where $\mathbf{x}$ is the output vector, $\mathbf{I}$ is an identity matrix, and $\mathbf{u}$ stands for the summation vector which summarizes the elements in a matrix in a row-wise fashion. $\mathbf{p}$ is the vector of exports, and $\mathbf{f}$ is the final demand vector representing the sum of both domestic final demand and exports. The matrix $\mathbf{Y}$ gives the flows of final products

$$\mathbf{Y} = \begin{bmatrix} \mathbf{Y}_{ss}^D & \cdots & \mathbf{Y}_{sr}^D \\ \mathbf{Y}_{ss}^H & \cdots & \mathbf{Y}_{sr}^H \\ \mathbf{Y}_{ss}^F & \cdots & \mathbf{Y}_{sr}^F \\ \vdots & \vdots & \vdots \\ \mathbf{Y}_{rs}^D & \cdots & \mathbf{Y}_{rr}^D \\ \mathbf{Y}_{rs}^H & \cdots & \mathbf{Y}_{rr}^H \\ \mathbf{Y}_{rs}^F & \cdots & \mathbf{Y}_{rr}^F \end{bmatrix}.$$

Its submatrices $\mathbf{Y}_{sr}^D$, $\mathbf{Y}_{sr}^H$, and $\mathbf{Y}_{sr}^F$ give the deliveries from mainland China-, HMT-, and foreign-owned enterprises in region $s(=1, \cdots, 31)$ for final demand in region $r(=1, \cdots, 31)$, respectively. We denote $\mathbf{A}$ as the interprovincial direct input-output coefficient matrix

$$\mathbf{A} = \begin{bmatrix} \mathbf{A}_{ss}^{DD} & \mathbf{A}_{ss}^{DH} & \mathbf{A}_{ss}^{DF} & \cdots & \mathbf{A}_{sr}^{DD} & \mathbf{A}_{sr}^{DH} & \mathbf{A}_{sr}^{DF} \\ \mathbf{A}_{ss}^{HD} & \mathbf{A}_{ss}^{HH} & \mathbf{A}_{ss}^{HF} & \cdots & \mathbf{A}_{sr}^{HD} & \mathbf{A}_{sr}^{HH} & \mathbf{A}_{sr}^{HF} \\ \mathbf{A}_{ss}^{FD} & \mathbf{A}_{ss}^{FH} & \mathbf{A}_{ss}^{FF} & \cdots & \mathbf{A}_{sr}^{FD} & \mathbf{A}_{sr}^{FH} & \mathbf{A}_{sr}^{FF} \\ \vdots & \vdots & \vdots & \vdots & \vdots & \vdots & \vdots \\ \mathbf{A}_{rs}^{DD} & \mathbf{A}_{rs}^{DH} & \mathbf{A}_{rs}^{DF} & \cdots & \mathbf{A}_{rr}^{DD} & \mathbf{A}_{rr}^{DH} & \mathbf{A}_{rr}^{DF} \\ \mathbf{A}_{rs}^{HD} & \mathbf{A}_{rs}^{HH} & \mathbf{A}_{rs}^{HF} & \cdots & \mathbf{A}_{rr}^{HD} & \mathbf{A}_{rr}^{HH} & \mathbf{A}_{rr}^{HF} \\ \mathbf{A}_{rs}^{FD} & \mathbf{A}_{rs}^{FH} & \mathbf{A}_{rs}^{FF} & \cdots & \mathbf{A}_{rr}^{FD} & \mathbf{A}_{rr}^{FH} & \mathbf{A}_{rr}^{FF} \end{bmatrix}$$

Its typical element $a_{sr}^{DF}(ij)$ provides the intermediate input from mainland China-owned enterprises of sector $i(=1, \cdots, n)$ in region $s$ used by foreign-owned enterprises of sector $j(=1, \cdots, n)$ in region $r$ for producing one unit of output.

Let $\mathbf{v}\prime = \mathbf{w}\prime(\hat{\mathbf{y}})^{-1}$ be the value-added coefficient vector, the elements of which provide the value-added per unit of output. Then the value-added vector $\mathbf{w}\prime$ can be written as

$$\mathbf{w}'\mathbf{u} = \mathbf{v}'(\mathbf{I} - \mathbf{A})^{-1}\mathbf{f} \tag{2}$$

Equation (2) calculates the summation of the value-added generated by all enterprises across all region-industry pairs in mainland China. We can attain the value-added in a specific region $r$ by replacing the vector $\mathbf{v}$ with $\mathbf{v}_r$. The new vector is of equal length, but all its elements except the value-added coefficients for industries in region $r$ are set to be zero. This yields

$$\mathbf{w}_r = (\mathbf{v}_r)\prime(\mathbf{I} - \mathbf{A})^{-1}\mathbf{f} \tag{3}$$

We can also obtain industry-level results by replacing $\mathbf{v}_r$ with $\hat{\mathbf{v}}_r$, in which circumflex denotes a diagonalization of the vector. In a similar way, we define $\mathbf{e}$ as the direct carbon emission coefficient vector. Then

we can calculate the carbon emission in region $r$ as

$$c_r = (\mathbf{e}_r)'(\mathbf{I} - \mathbf{A})^{-1}\mathbf{f} \qquad (4)$$

There are two methods to calculate the contribution of MNEs: the decomposition method[48,49] based on IO model and the IO-based hypothetical extraction method (HEM)[50–52]. The two methods yield the same results (see Supplementary Note 3 for the mathematical proof). However, Los et al.[50] note that the decomposition method is overly complex in terms of both mathematical derivation and interpretation and propose that the more intuitive HEM can be more easily applied to the data. The HEM has been widely employed to quantify the importance of specific industries or enterprises to an economy[10,53]. The core idea of this technique is to measure how much the economic output of an economy would be affected assuming a particular set of industries were hypothetically removed. Here, we present how to calculate the value-added in region $r$ that are generated by foreign-owned enterprises using the HEM under the assumption that the absence of a certain type of enterprise in an industry would not lead to substitution in the short run[50]. The value-added under HEM can be written as

$$\widetilde{\mathbf{w}_r} = (\mathbf{v}_r)'\left(\mathbf{I} - \widetilde{\mathbf{A}}\right)^{-1}(\widetilde{\mathbf{Y}}\mathbf{u} + \widetilde{\mathbf{p}}) \qquad (5)$$

where

$$\widetilde{\mathbf{A}} = \begin{bmatrix} \mathbf{A}_{ss}^{DD} & \mathbf{A}_{ss}^{DH} & \mathbf{O} & \cdots & \mathbf{A}_{sr}^{DD} & \mathbf{A}_{sr}^{DH} & \mathbf{O} \\ \mathbf{A}_{ss}^{HD} & \mathbf{A}_{ss}^{HH} & \mathbf{O} & \cdots & \mathbf{A}_{sr}^{HD} & \mathbf{A}_{sr}^{HH} & \mathbf{O} \\ \mathbf{O} & \mathbf{O} & \mathbf{O} & \cdots & \mathbf{O} & \mathbf{O} & \mathbf{O} \\ \vdots & \vdots & \vdots & \vdots & \vdots & \vdots & \vdots \\ \mathbf{A}_{rs}^{DD} & \mathbf{A}_{rs}^{DH} & \mathbf{O} & \cdots & \mathbf{A}_{rr}^{DD} & \mathbf{A}_{rr}^{DH} & \mathbf{O} \\ \mathbf{A}_{rs}^{HD} & \mathbf{A}_{rs}^{HH} & \mathbf{O} & \cdots & \mathbf{A}_{rr}^{HD} & \mathbf{A}_{rr}^{HH} & \mathbf{O} \\ \mathbf{O} & \mathbf{O} & \mathbf{O} & \cdots & \mathbf{O} & \mathbf{O} & \mathbf{O} \end{bmatrix}$$

$$\widetilde{\mathbf{Y}} = \begin{bmatrix} \mathbf{Y}_{ss}^{D} & \cdots & \mathbf{Y}_{sr}^{D} \\ \mathbf{Y}_{ss}^{H} & \cdots & \mathbf{Y}_{sr}^{H} \\ \mathbf{O} & \cdots & \mathbf{O} \\ \vdots & \vdots & \vdots \\ \mathbf{Y}_{rs}^{D} & \cdots & \mathbf{Y}_{rr}^{D} \\ \mathbf{Y}_{rs}^{H} & \cdots & \mathbf{Y}_{rr}^{H} \\ \mathbf{O} & \cdots & \mathbf{O} \end{bmatrix}, \widetilde{\mathbf{p}} = \begin{bmatrix} \mathbf{p}_{s}^{D} \\ \mathbf{p}_{s}^{H} \\ \mathbf{O} \\ \vdots \\ \mathbf{p}_{r}^{D} \\ \mathbf{p}_{r}^{H} \\ \mathbf{O} \end{bmatrix}$$

and $\mathbf{O}$ denotes a matrix of appropriate dimension filled with zeros. The difference between the original value-added and the value-added under HEM is the contribution of foreign-owned enterprises. The contribution of HMT-owned enterprises can be calculated in a similar way. The same applies to the calculation of carbon emissions in region $r$ that are generated by foreign-owned enterprises after replacing $\mathbf{v}_r$ with $\mathbf{e}_r$.

## Carbon Gini coefficient

The carbon Gini (C-Gini) coefficient is constructed by adapting the well-known income Gini coefficient, which was originally developed by Italian statistician Corrado Gini to measure the inequality among different income levels[54] and now has been widely adapted to measure the inequality of any distribution[55,56]. The income Gini coefficient is built on the comparison of the cumulative proportions of the population against the cumulative proportions of income they receive. If a small proportion of the population possesses all the income while the rest has none, it results in extreme inequality. Here, we aim to assess the inequality in $CO_2$ emissions across regions relative to their value-added gains. If regions with minimal value-added emit the majority of

the $CO_2$ emissions, this also represents extreme inequality. Therefore, we construct the C-Gini by comparing the cumulative proportions of value-added against the cumulative proportions of $CO_2$ the regions emit[24,57]:

$$G = \sum_{k=1}^{m} D_k Y_k + 2\sum_{k=1}^{m} D_k(1 - T_k) - 1 \qquad (6)$$

where G is the C-Gini coefficient. $D_k$ and $Y_k$ are the proportions of the value-added and carbon emissions of group $k(=1, \cdots, m)$, respectively. $T_k$ is the cumulative proportion of carbon emissions across group 1 until group $k$. There are 30 groups (provincial units) in our calculation.

We calculate both an original and a hypothetical Gini coefficient based on production-based emissions. The original C-Gini (G) is derived from the original provincial carbon emissions and value-added, while the hypothetical C-Gini ($\widetilde{G}$) is calculated using the hypothetical values of provincial carbon emissions and value-added which are provided by Eq. (5). The difference ($\hat{G}$) between them gives the contribution of MNEs to the C-Gini coefficient

$$\hat{G} = \widetilde{G} - G. \qquad (7)$$

We perform a Diebold-Mariano (DM) test to determine whether the original and hypothetical C-Ginis differ significantly. This test evaluates if the differences in outcomes between the two models are statistically meaningful rather than merely resulting from random fluctuations. We calculate both the original and hypothetical C-Ginis for China as a whole, as well as for its four zones across five years. Using this sample, we conduct the DM test, and the results confirm that these differences are statistically significant.

## Structural decomposition analysis

We employ an SDA to attribute the role of different driving factors in the changes in the MNEs contribution to the C-Gini between time periods. It is clear from Eq. (7) that the value of $\hat{G}$ depends on the direct carbon emission coefficient $\mathbf{e}$, the value-added coefficient $\mathbf{v}$, the input matrix $\mathbf{A}$, and the final demand $\mathbf{f}$. The matrix $\mathbf{A}$ is split into a part that gives the overall production technology ($\mathbf{A}^*$), reflecting the total direct inputs that are required per unit of output, and the other part that gives the regional shares ($\boldsymbol{\Phi}$), which captures the structure of inter-regional intermediate inputs[58]. The matrix $\mathbf{A}^*$ is defined as

$$\mathbf{A}^* = \begin{bmatrix} \mathbf{A}_{*1}^{DD} & \mathbf{A}_{*1}^{DH} & \mathbf{A}_{*1}^{DF} & \cdots & \mathbf{A}_{*r}^{DD} & \mathbf{A}_{*r}^{DH} & \mathbf{A}_{*r}^{DF} \\ \mathbf{A}_{*1}^{HD} & \mathbf{A}_{*1}^{HH} & \mathbf{A}_{*1}^{HF} & \cdots & \mathbf{A}_{*r}^{HD} & \mathbf{A}_{*r}^{HH} & \mathbf{A}_{*r}^{HF} \\ \mathbf{A}_{*1}^{FD} & \mathbf{A}_{*1}^{FH} & \mathbf{A}_{*1}^{FF} & \cdots & \mathbf{A}_{*r}^{FD} & \mathbf{A}_{*r}^{FH} & \mathbf{A}_{*r}^{FF} \\ \vdots & \vdots & \vdots & \vdots & \vdots & \vdots & \vdots \\ \mathbf{A}_{*1}^{DD} & \mathbf{A}_{*1}^{DH} & \mathbf{A}_{*1}^{DF} & \cdots & \mathbf{A}_{*r}^{DD} & \mathbf{A}_{*r}^{DH} & \mathbf{A}_{*r}^{DF} \\ \mathbf{A}_{*1}^{HD} & \mathbf{A}_{*1}^{HH} & \mathbf{A}_{*1}^{HF} & \cdots & \mathbf{A}_{*r}^{HD} & \mathbf{A}_{*r}^{HH} & \mathbf{A}_{*r}^{HF} \\ \mathbf{A}_{*1}^{FD} & \mathbf{A}_{*1}^{FH} & \mathbf{A}_{*1}^{FF} & \cdots & \mathbf{A}_{*r}^{FD} & \mathbf{A}_{*r}^{FH} & \mathbf{A}_{*r}^{FF} \end{bmatrix}$$

with the sub-matrix $\begin{bmatrix} \mathbf{A}_{*r}^{DD} & \mathbf{A}_{*r}^{DH} & \mathbf{A}_{*r}^{DF} \\ \mathbf{A}_{*r}^{HD} & \mathbf{A}_{*r}^{HH} & \mathbf{A}_{*r}^{HF} \\ \mathbf{A}_{*r}^{FD} & \mathbf{A}_{*r}^{FH} & \mathbf{A}_{*r}^{FF} \end{bmatrix} = \begin{bmatrix} \sum_{s=1}^{31}\mathbf{A}_{sr}^{DD} & \sum_{s=1}^{31}\mathbf{A}_{sr}^{DH} & \sum_{s=1}^{31}\mathbf{A}_{sr}^{DF} \\ \sum_{s=1}^{31}\mathbf{A}_{sr}^{HD} & \sum_{s=1}^{31}\mathbf{A}_{sr}^{HH} & \sum_{s=1}^{31}\mathbf{A}_{sr}^{HF} \\ \sum_{s=1}^{31}\mathbf{A}_{sr}^{FD} & \sum_{s=1}^{31}\mathbf{A}_{sr}^{FH} & \sum_{s=1}^{31}\mathbf{A}_{sr}^{FF} \end{bmatrix}$. These matrices give the aggregated amounts of intermediate inputs per unit of output irrespective of their geographical sources. The matrix $\boldsymbol{\Phi}$ is given by $\Phi = A//A^*$, where $//$ denotes element-wise division. Similarly, the final demand ($\mathbf{f}$) can be split into the total final demands ($\mathbf{f}^*$) and the regional structure ($\boldsymbol{\Pi}$) of final demands.

The value of Ĝ can now be written as a function of six matrices and vectors: $\hat{G} = f(\mathbf{e}, \mathbf{v}, \mathbf{A}^*, \boldsymbol{\Phi}, \mathbf{f}^*, \boldsymbol{\Pi})$. The changes in Ĝ can be decomposed into the contributions of six driving factors using the average of two polar decompositions[22]:

$$\Delta\hat{G} = \Delta\mathbf{e} + \Delta\mathbf{v} + \Delta\mathbf{A}^* + \Delta\boldsymbol{\Phi} + \Delta\mathbf{f}^* + \qquad (8)$$

where $\Delta\mathbf{e}$, $\Delta\mathbf{v}$, $\Delta\mathbf{A}^*$, $\Delta\boldsymbol{\Phi}$, $\Delta\mathbf{f}^*$, and $\Delta\boldsymbol{\Pi}$ give the contribution of the direct carbon emission coefficient, the value-added coefficient, the overall production technology, the structure of inter-regional intermediate inputs, the total final demands, and the regional structure of final demands, respectively. The changes in the geographical distribution of MNEs are reflected in both $\Delta\boldsymbol{\Phi}$ and $\Delta\boldsymbol{\Pi}$, while $\Delta\mathbf{A}^*$ and $\Delta\mathbf{f}^*$ also capture the changes in the structures of intermediate products and final products provided by MNEs, respectively.

## Limitations

One limitation of this study may stem from potential uncertainties within the province-industry $CO_2$ emission inventory, which is sourced from the CEADs database. According to the database developers[59,60], several factors can introduce different levels of uncertainty into the emission data, including activity data, emission factors, data completeness, and measurement errors. To address these issues, they employ Monte Carlo simulation to assess data uncertainty and update the inventory as new information becomes available. Despite these uncertainties, their estimates are regarded as reliable and are frequently used by researchers.

A second limitation may arise from the scenario analysis. Our econometric regression analysis revealed that the knowledge spillover effects of MNEs enable technological convergence, allowing the emission technology of lagging enterprises to catch up with that of the leaders. Based on this solid empirical evidence, we designed four distinct scenarios with a comprehensive consideration of practicality and feasibility, considering the potential upgrades in emission technology and adjusting the emission coefficients of technological laggards across different regions and industries. Meanwhile, we acknowledge that the upgrading of emission technology may also increase value-added creation capacity. In such a case, emission technology improvements in inland regions could reduce carbon inequality to a larger extent than our scenario estimates indicate. Unfortunately, we lack quantitative evidence to determine the extent to which changes in emission coefficients lead to variations in value-added creation. Such changes could alter economic structures and reshape interregional input-output linkages. Including these channels arbitrarily in the simulation would contradict the fundamental premise of the IPIO-based model and introduce substantial uncertainty to the results. Therefore, we have consciously chosen not to include changes in the value-added coefficient or the input-output structure in our simulation and leave it for further exploration in future work.

## Data availability

This study uses an interprovincial input-output (IPIO) dataset that distinguishes three types of enterprise ownership for mainland China[14] (https://www.nature.com/articles/s41597-023-02183-2). The table covers 31 provincial units of mainland China and 42 sectors in each province. The sector classifications are not consistent in the tables for different years. We thus aggregate some sectors and derive 30 sectors (Supplementary Table 2). The sectoral $CO_2$ emission inventories are obtained from the CEADs database[59,60] (https://www.ceads.net/). CEADs provides sectoral emission data for 30 provinces except Tibet. We thus include 30 provinces in our calculation. However, CEADs do not distinguish emissions from domestic-owned, HMT-owned or foreign-owned enterprises. This study adopts the approach of Zhang

et al.[7] and disaggregates sectoral emissions into three parts according to the intermediate use of fuel resources of the three types of enterprises, which are provided by the IPIO table. More detailed descriptions of the data are provided in Supplementary Note 1.

## Code availability

The relevant R codes are publicly available at Zenodo (https://doi.org/10.5281/zenodo.15489579).

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

## Acknowledgements

The authors gratefully acknowledge the financial support from the National Natural Science Foundation of China (grant Nos. 72373141 (K.T.), 71903186 (K.T.), 72403115 (Y.Z.), 71988101 (C.Y.), and 72074208 (K.T.)), Major Programme of National Fund of Philosophy and Social Science of China (grant No. 19ZDA062 (C.Y.)), TPESER Youth Innovation Key Programme (grant No. TPESER-QNCX2022ZD-04 (Z.Z.)), and the European Union (grant No. 101137905 (J.M.)).

## Author contributions

K.T. designed the research, determined the methods, and contributed to interpreting the outcomes and writing the manuscript. Y.Z. contributed to the calculation and interpretation of the outcomes. Z.Z. and J.M. contributed to interpreting the outcomes and writing. Y.S. contributed to the writing and plotted the figures. C.Y., H.Z., and X.N. improved the analysis.

## Competing interests

The authors declare no competing interests.
