## [Transparent Peer Review file · Nature Communications]

Leveraging multinational enterprises to reduce the escalating regional carbon inequality in China

Corresponding Author: Professor Jing Meng

Version 0:

Reviewer comments:

Reviewer #1

(Remarks to the Author)

This paper first estimates the carbon emissions of domestic enterprises and MNEs and use Gini index and hypothetical Gini index to assess the impacts of MNEs on China's regional carbon emission inequality. There has been some studies on international and inter-regional carbon imbalance and inequalities, and this paper adds to this line of literature by considering the role of MNEs. This paper finds that reducing the carbon intensity to the same level as MNEs could reduce carbon emissions and carbon inequality at the same time. Overall, this paper is clear and well organized. However, there are several concerns that need to be addressed.

1. Method. The role of MNEs in the Gini index.

-The manuscript seems to quantify the role of MNEs on the Gini coefficient through simple addition and subtraction method (e.g., from 0.22 to 0.23 in 2002 is determined as an 4.3% increase). This method equates to an assumption about the structure of the input-output model, specifically, that the absence of a certain type of company in an industry would not lead to changes/fluctuations in carbon emissions of domestic firms—no substitution. If this is the case, such assumption should be explicitly stated and justified.

-Is it feasible to compare different Gini coefficients directly by subtracting them to assess the impact of specific components, as what has been adopted in this study (e.g., comparing 0.22 to 0.23 in 2002)? This approach could be likened to calculating the Gini coefficient with and without a specific entity (e.g., an individual's income) to gauge their influence. The validity of this method and its mathematical justification need clear explanation in the method section.

-The results of the paper suggest a minor impact of MNEs on the Gini coefficient, with changes as minimal as from 0.22 to 0.23 in 2002, or from 0.23 to 0.25 to 2007. It is crucial to determine whether these changes are statistically significant. The manuscript would benefit from statistical tests to substantiate the claimed impact of MNEs on changes in carbon inequality from 2002-2012 and a decline in 2017.

-The manuscript currently calculates the Gini coefficient using a single indicator (e.g., 0.22, 0.23, 0.25), which does not differentiate the contributions from different components, such as domestic enterprises, Hong Kong, Macau, and Taiwan enterprises, or foreign enterprises. This aggregation limits the ability to substantiate claims regarding the impact of MNEs on inequality thoroughly. It would be beneficial to employ a decomposition method to quantify the individual contributions of each type of enterprise to the Gini coefficient, thus providing a more nuanced understanding of the sources of inequality.

-The specific role of MNEs in carbon inequality levels seems the central theme of the article. Thus, it would be better to use some quantitative analysis to support the narrative of MNEs' role in inequality in China. Including more detailed charts and tables would strengthen this part of the study and provide clearer evidence.

2. Method. The manuscript employs regression analysis to examine the role of MNEs in reducing carbon emission intensity.

There are two set of dependent variables: (1) industry emission intensity; (2) the ratio of domestic firm emission intensity to MNE emission intensity. The independent variable considered is the proportion of MNE intermediate inputs in an industry. The logic here is that: (1) if industries utilizing more MNE inputs demonstrate lower emission intensities, it suggests that MNEs can help reduce emissions; (2) if there are narrower gaps in emission intensity between domestic and foreign firms when more MNE intermediate inputs are used, it implies MNEs can lessen emission intensity gaps. However, this analysis might have conflated correlation with causality, and the relationship observed could be influenced by unaccounted variables, such as industry-specific technologies or regulatory environments.

-The current findings in the manuscript could indeed be influenced by endogeneity and reverse causality, which complicate the interpretation of the effects of MNEs on emission intensity. For instance, industries with inherently lower emission

intensities might naturally attract more MNEs due to factors such as alignment with global environmental standards or the industry's higher technology and service-oriented focus. These industries, like service sectors or high-tech industries, typically have lower emissions compared to more traditional, labor-intensive industries. Additionally, in such sectors, the gap in emission intensity between domestic and foreign firms might already be minimal due to the nature of the industry itself, rather than the influence of MNEs. Therefore, without addressing these potential endogeneity issues through robust statistical methods like instrumental variables or fixed effects models, the regression results might overstate the impact of MNE inputs on reducing emission disparities.

-The manuscript discusses two potential mechanisms by which MNEs could influence a country's emission intensities. However, the current regression analysis does not adequately determine which mechanism dominates. The two impacts mentioned as 'the pollution haven effect'—increasing the emission coefficient—and 'the knowledge spillover effect'—leading to technological upgrading that reduces the emission coefficient cannot be justified or ruled out based on the above regression. For instance, if MNEs contribute to a "pollution haven" effect, where domestic firms increase emissions in sectors that are more upstream (which may use fewer MNE intermediates), the findings could still align with the observed results: those who use less MNEs inputs can have a higher emission intensity. However, this is not induced by the 'knowledge spillover effect'. Instead, it is due to the contrary effect. Thus, the current regression setup cannot provide enough evidence to distinguish between these mechanisms.

-In assessing the robustness of the regression analysis in your manuscript, it's essential to consider whether robust error terms are being used. If robust errors are not being applied, possible violations of regression assumptions such as heteroscedasticity or autocorrelation may bias standard error estimates and influence the significance levels of coefficients. Clustering standard errors at appropriate levels, such as industry or regional levels, can further enhance the analysis. This is particularly important when the data is across different industries or geographical areas, to control for unobserved heterogeneity within these groups.

-Temporal and Spatial Autocorrelation.

-Please report the number of observations as well.

3. Method. In the scenario analysis, the current method approach involves directly reducing the emission intensity of the highest emitters. For instance, scenario 1 directly assumes that the domestic enterprises in the western regions are to have the same emission intensities of MNEs. This can lead to a straightforward result—lower inequality levels, as it just flattens the Lorenz curve.

-However, considering an alternative approach to "leveraging MNEs" that involves further lowering the emission intensities of provinces on the left of the Lorenz curve could present different outcomes. For example, if the scenario assumes that domestic enterprises in the eastern regions have the same emission intensities of their own MNEs, it might actually increase inequality.

-Why the certain scenarios in the manuscript are chosen instead of the alternative proposed? It would be better to have some justifications to support the scenario settings.

4. Method. The hypothetical extraction method (HEM) is chosen instead of the decomposition method.

-Why HEM is chosen over the decomposition method? Are there any particular advantages of choosing HEM over decomposition?

-Although it is stated here that the HEM and decomposition methods would yield consistent results, are there any mathematical proof for this?

-How does this study deal with this? Under the HEM, emissions are regarded as related to domestic firms/HMT firms/foreign firms. However, if I understand correctly, there are some parts of emissions that would be quantified as related to HMT firms and as related to foreign firms as well. For instance, considering the simplest case that there is only 1 region, 1 sector, and 2 types of firms.

$A = | \text{Add Adf, Afd Aff} |$

$A' = | \text{Add 0, 0 0} |$

$A'' = | 0 0, 0 \text{Aff} |$

$F = | \text{Fdd Fdf, Ffd Fff} |$

$F' = | \text{Fdd 0, 0 0} |$

$F'' = | 0 0, 0 \text{Fff} |$

So, if the total output is represented by AF, and the portion unrelated to foreign investment is A'F', will subtracting A'F' from AF result in the foreign-influenced part? If the portion unrelated to domestic investment is A''F'', will subtracting A''F'' from AF result in the domestic-related part? Do these two components added together still equal the total output?

5. Results. Figure 4 states that 'China has undergone an escalating regional carbon inequality during 2002–2017, as evidenced by the rise in its original C-Gini coefficient from 0.23 in 2002, to 0.25 in 2007, 0.28 in 2012, and 0.32 in 2017.'

-Is the C-Gini based on consumption-based emissions, production-based emissions, or the HEM method emissions?

-This study finds an increase in carbon inequality. How does this compare with other similar studies on regional carbon inequalities? Can this finding be reconciled with previous studies that observed a decrease in inequality? How can the differences be explained, e.g., are they due to the differences between the residential sector and overall final demand, or are there any other factors?

Minor

-As China's entry into the WTO happened in 2001, can the period from 2002 to 2017 fully capture the rapid expansion of foreign investment and MNEs? Given that an inter-provincial IO table for China has already been established covering the long period from 1987 to 2017, it might be worthwhile to extend the analysis to include more years. Including data from 1997 as a comparative benchmark might provide additional insights.

-Page 5 line 125, Figure 1. It would be better if both Figure 1 a and b use the same color legend to represent the same item

for clarity and consistency.

-Page 7 line 167-170. The statement that 'from 2002 to 2017, there was a distinct shift in the geographical distribution of value-added generated by MNEs, marking a transition from the east to the west' might be reconsidered. The change from 71.2% to 67.6% over fifteen years could be perceived as relatively minor. It may be more accurate to describe this shift as gradual rather than 'distinct' to reflect the modest scale of change more precisely.

-Page 11 line 249-250. The statement that 'The exacerbated carbon inequality over the 2002–2017 period is determined by changes in the cumulative proportions of CO₂ emissions and value-added' seems redundant. This is because by definition the Gini coefficient inherently measures inequality based on cumulative proportions, describing this as a novel finding may not be necessary. It would be more insightful to contextualize how these specific changes contribute uniquely to carbon inequality or to explore additional factors that might have influenced the trends observed during this period.

-Page 11, lines 260-265. Although line 249-250 shows that 'from 2002 to 2017, the eastern regions experienced a faster economic development', while the provided example here argues that Ningxia (a western region) experienced a 736.4% increase in value-added, compared to Guangdong's (an eastern region) 564.2% increase. This seems to contradict the general statement about regional economic growth rates. To avoid confusion and strengthen the argument, it might be beneficial to select examples that align more closely with the initial claim about regional growth patterns, or to clarify why these particular examples were chosen despite the apparent discrepancy.

-P23 Equation 6. Since this manuscript specifically addresses carbon inequality, it would be more appropriate to replace the equation for the Gini coefficient of income inequality with one that directly can be applied to estimate Chinese carbon inequality.

(Remarks on code availability)

No code shared at this stage.

Reviewer #2

(Remarks to the Author)

This study focused on China and uncovered the relationships between its exacerbated carbon inequality and MNEs. While this kind of work is necessary, I believe that this manuscript does not meet the publication criteria of Nature Communications for the following reasons.

1.The research background section puts forward the scientific problem of this research more clearly, but the sorting out of existing research is weak, and the discussion of whether this research has important theoretical value or application prospects is insufficient.

2.The introduction does not explain whether there is any innovation in theory, method or technology in this research, and the innovation of the research is insufficient. What is the significance of quantitative research on the impact of MNEs on carbon inequality?

3.Does the innovation of this paper only increase the impact of MNEs on carbon inequality compared to the following studies?

Zhang, W., Liu, Y., Feng, K., Hubacek, K., Wang, J., Liu, M., Jiang, L., Jiang, H., Liu, N., Zhang, P., Zhou, Y., Bi, J., 2018a. Revealing Environmental Inequality Hidden in China's Inter-regional Trade. *Environ. Sci. Technol.* 52, 7171–7181.

Zhang, H., Zhang, W., Lu, Y., Wang, Y., Shan, Y., Ping, L., Li, H., Lee, L.-C., Wang, T., Liang, C., Jiang, H., Cao, D., 2023. Worsening Carbon Inequality Embodied in Trade within China. *Environ. Sci. Technol.* 57, 863–873.

4.The research methodology of this paper is more reasonable and can basically solve the scientific problem of this paper. However, there is no mention of the advantages of using the Gini coefficient to characterize carbon inequality, nor is interregional carbon inequality taken into account.

5.In measuring carbon inequality (C - Gini), this paper replace the income with carbon emissions and the population with value-added, please explain its rationality.

6.The resolution of Figure 1 is low, and it is not clear and easy to read.

7.It is suggested to abbreviate the names of provinces in Fig3-Fig5.

8.It is mentioned in the part of result analysis that reducing carbon emission coefficient can reduce carbon inequality, but reducing carbon emission coefficient can only reduce carbon emission to a certain extent. How to further reduce carbon inequality needs to be analyzed in combination with the transfer of value-added.

9.There are many factors affecting carbon inequality. On the whole, the research content of this paper is insufficient and lacks depth. It is suggested to explore the deep causes of carbon inequality changes.

10.The policy recommendations for reducing carbon inequality in the discussion appeared to be too general.

11.What are the limitations of this study and future work? They should be supplemented.

12.Note that references should be formatted consistently.

(Remarks on code availability)

Version 1:

Reviewer comments:

Reviewer #1

(Remarks to the Author)

After reading the paper and the authors' reply thoroughly, I am satisfied that they have addressed the issues raised in previous version. My recommendation is therefore for the paper to be accepted for publication with minor revisions.

- Figure 2. The numbers in the circles are difficult to read. Please either increase the font size or use a different color.
- Figure 3. In "value-added/CO2 emissions", please check whether "and" should be used. Also, please delete extra " " in the caption.
- Figures 4 and 5, remove the extra " " in the caption.
- Please move the DM (Diebold-Mariano) test section from the supplementary file to the method section.
- Please add a section about the limitations in the method section, which have already been discussed in your response letter.

(Remarks on code availability)

Reviewer #2

(Remarks to the Author)

Thanks to the author's response, my questions have been answered. However, I still believe that this manuscript does not meet the publication criteria of Nature Communications for the following reasons:

- 1.Although the authors emphasize the novelty and contributions of this paper, I still believe that on the basis of existing research, this article does not represent a significant breakthrough.
- 2.The depth and breadth of this study is insufficient and does not significantly expand or deepen existing research.
- 3.Changes in emission coefficients lead to changes in value added, which in turn leads to changes in carbon inequality. However, in the scenario setting, the authors did not consider the change in value added and considered it beyond the scope of the study. Therefore, the rationality and accuracy of the final carbon inequality cannot be guaranteed.
- 4.This paper only considers the overall carbon inequality at the regional level, lacking an examination of the impact of multinational enterprises on the flow of carbon emissions and value added between regions.
- 5.With the existing quality of this paper, I think it is more suitable for journals such as 《Journal of Cleaner Production》 and 《Environmental Science & Technology》, etc.

(Remarks on code availability)

No code shared

Reviewer #3

(Remarks to the Author)

The authors studied the effects of MNEs on regional carbon inequality in China. Although I find the topic a trendy one (sustainability), I hope the comments below could help improve/reposition the paper.

Some statements are strong but unjustified. For example, "The presence of MNEs increased China's carbon inequality from 1997 to 2012." It could be a mere correlation as the HEM exercises do not guarantee a ceteris paribus interpretation.

The instrumental variable (FMA) is a problematic one. Distance to the nearest port is far from random in this context, but is highly correlated with East/West or Coastal/Inland divide in China. A simple balance check could reveal this problem.

What happened in (or around) 2017? 2017 has been highlighted throughout the paper as a year of pivotal change. But what exactly happened to the MNEs in China? Is this in line with the mechanism of carbon reduction? If so, any macro or micro level evidence?

The policy implications are limited (perhaps due to the descriptive results and the complex factors embedded into "MNEs"). The word "leveraging" in the title implies as if MNEs could be an exogenous force subject to policy manipulation. Unfortunately, the regional disparities in terms of natural and human resources are likely to be responsible for both carbon inequality and MNEs choices.

Further proofreading:

For example, the authors state, on page 11, that "However, the economic growth rate in the west did not match the pace of emission growth". However, the mismatch was observed in the east (e.g., Jiangsu with high economic growth but moderate emission growth).

(Remarks on code availability)

Version 2:

Reviewer comments:

Reviewer #1

(Remarks to the Author)

The authors have further revised to increase the readability of the figures and include more information on the methods and limitations. The author has responded well to all my comments. I would also like to thank the authors for their efforts.

Reviewer #2

(Remarks to the Author)

At this revised manuscript authors developed a substantially improved study, comparing to the initial study, having the review comments addressed in a systematic and responsible manner. My recommendation is therefore for the paper to be accepted for publication.

Reviewer #3

(Remarks to the Author)

While the authors' efforts to revise the paper are appreciated, several concerns remain.

1. Central Issue

The argument that MNEs are a driving factor behind carbon inequality is unconvincing. MNEs might not be an appropriate focal variable for deriving meaningful policy implications.

2. Econometric Concerns

There are several econometric issues that need to be addressed:

Many important covariates are not controlled for, such as urbanization, administrative quality, and other potential factors.

The IV "FMA" is likely correlated with these omitted covariates, meaning it may influence the dependent variable (emissions) through channels other than "MNEs". This violates the "exogeneity" condition.

Consider replacing "MNEs" with an unrelated variable, such as "number of foreign tourists". The IV test might still be passed, and significant results might still be obtained. However, it would be unreasonable to suggest mitigating carbon inequality by increasing the number of foreign tourists.

3. Policy Implications

If MNEs are not an exogenous force, the validity of policy recommendations based on MNEs is questionable. Mitigating emission inequality might be more effectively achieved by promoting low-carbon technologies and enforcing stringent CO₂ mitigation policies for domestic enterprises (mentioned by the authors), without involving MNEs.

4. The Role of 2017

It remains unclear why the year 2017 is considered pivotal:

Was the observed increase in the proportion of MNEs located in Western China due to more MNEs relocating to this region, or was it the result of coastal MNEs moving abroad?

Development plans such as the Western Development and the Rise of Central China began in the 2000s. Why would their effects only materialize around 2017?

The authors also mentioned the spillover effects of MNEs, but there seems to be no direct quantitative evidence supporting this mechanism. Also, if spillover effects are indeed a key mechanism, similar effects should be observable within domestic enterprises, at least in recent years.

Response to the Reviewers

From: “Leveraging multinational enterprises to reduce the escalating regional carbon inequality in China” (NCOMMS-24-21432)

The authors are grateful for the insightful comments and constructive suggestions from the reviewers. We have carefully revised the manuscript accordingly, which we believe has made for a better article. Please find below our detailed point-by-point responses to the reviewers. To enhance readability, we have copied the feedbacks of the reviewers and put our responses in italics.

REVIEWER COMMENTS

Reviewer #1 (Remarks to the Author):

This paper first estimates the carbon emissions of domestic enterprises and MNEs and use Gini index and hypothetical Gini index to assess the impacts of MNEs on China’s regional carbon emission inequality. There has been some studies on international and inter-regional carbon imbalance and inequalities, and this paper adds to this line of literature by considering the role of MNEs. This paper finds that reducing the carbon intensity to the same level as MNEs could reduce carbon emissions and carbon inequality at the same time. Overall, this paper is clear and well organized. However, there are several concerns that need to be addressed.

1. Method. The role of MNEs in the Gini index.

-The manuscript seems to quantify the role of MNEs on the Gini coefficient through simple addition and subtraction method (e.g., from 0.22 to 0.23 in 2002 is determined as an 4.3% increase). This method equates to an assumption about the structure of the input-output model, specifically, that the absence of a certain type of company in an industry would not lead to changes/fluctuations in carbon emissions of domestic firms—no substitution. If this is the case, such assumption should be explicitly stated and justified.

***Response:** Thank you for the comment. We have accordingly added a clear justification for the assumptions underlying our method (page 24). We employ the hypothetical extraction method (HEM), which is a widely used input-output (IO) based method, to account for the value-added and carbon emissions generated by MNEs. The traditional HEM by national IO tables usually assume that the extracted parts will be replaced by imports outside the system. Dietzenbacher et al. (2019) extend the national HEM to the world IO framework, indicating the extracted parts in a specific region can be replaced by products from other regions. However, Los et al. (2016) apply the HEM to the world IO model with the no-substitution assumption and derive results identical to those of related studies.*

There is no consensus on the necessity of considering substitution when using the HEM; it depends on the research question. The intermediate input matrix in the IO model reflects both production technology and economic linkages. In the case of footprint analysis which traces

economic benefits, emissions or materials embodied in a specific supply chain, the information on the economic linkages is fundamental for footprint analysis. Therefore, studies in footprint analysis, such as Los et al. (2016), Zhang et al. (2020), and Tian et al. (2022), believe that information on economic linkages is more important and assume no substitution when using the HEM. In the case of analyzing short-run impacts of disruptions or disasters, it is preferable to keep production technology constant whilst modifying economic linkages, since firms cannot change the production technology in the short run but may alter their products sources (Dietzenbacher et al., 2019). In this case, it is more appropriate to consider substitution. In this paper, we aim to account for the contribution (both economic and carbon footprints) of MNEs. Thus, we believe the no-substitution assumption is applicable for this study. We also mathematically prove that the HEM yields the same results as the decomposition method (Koopman et al., 2014), as demonstrated in Supplementary information 3.

*Dietzenbacher, E., van Burken, B. & Kondo, Y. (2019). Hypothetical extractions from a global perspective. *Economic Systems Research* 31, 505-519.*

*Koopman, R., Wang, Z., & Wei, S.J. (2014). Tracing value-added and double counting in gross exports. *American Economic Review*, 104(2), 459-494.*

*Los, B., Timmer, M. P., & de Vries, G. J. (2016). Tracing value-added and double counting in gross exports: Comment. *American Economic Review*, 106(7), 1958-1966.*

*Zhang, Z. et al. (2020). Embodied carbon emissions in the supply chains of multinational enterprises. *Nature Climate Change* 10, 1096-1101.*

*Tian, K. et al. (2022). Regional trade agreement burdens global carbon emissions mitigation. *Nature Communications* 13, 408.*

-Is it feasible to compare different Gini coefficients directly by subtracting them to assess the impact of specific components, as what has been adopted in this study (e.g., comparing 0.22 to 0.23 in 2002)? This approach could be likened to calculating the Gini coefficient with and without a specific entity (e.g., an individual's income) to gauge their influence. The validity of this method and its mathematical justification need clear explanation in the method section.

Response: *Thanks for your suggestion. We have added a clear explanation in the method section with regards to the validity of our method and the mathematical justification (page 24 and the Supplementary information 3). We compare two distinct C-Gini coefficients: the original C-Gini which is derived from provincial value-added and CO₂ emissions in their original form, and the hypothetical C-Gini which is computed using hypothetical values of provincial value-added and CO₂ emissions. The hypothetical values are obtained by extracting the contribution of MNEs. This idea follows the hypothetical extraction method (HEM). It is valid, widely accepted, and broadly adopted to quantify the importance of specific industries or enterprises to an economy. The core idea of this technique is to measure how much the economic output of an economy would be affected assuming a particular set of industries were hypothetically removed.*

To give a concrete example, when calculating the domestic value added in exports, Los et al. (2016) compare the actual value added with the hypothetical value added which is computed by extracting export flows (setting exports to zero). They define the difference between the two values

as the contribution of exports to value added, i.e., the domestic value added in exports. They prove that the HEM derives the same results as Koopman et al. (2014)'s decomposition method. For additional examples, employing a similar idea, Chen et al. (2018) assess the economic exposures to Brexit, and Tian et al. (2023) evaluate the economic exposures to regional value chain disruptions due to the Covid lockdown.

Koopman, R., Wang, Z., & Wei, S.J. (2014). Tracing value-added and double counting in gross exports. American Economic Review, 104(2), 459-494.

Los, B., Timmer, M. P., & de Vries, G. J. (2016). Tracing value-added and double counting in gross exports: Comment. American Economic Review, 106(7), 1958-1966.

Chen, W., Los, B., McCann, P., Ortega - Argilés, R., Thissen, M., & van Oort, F. (2018). The continental divide? Economic exposure to Brexit in regions and countries on both sides of The Channel. Papers in Regional Science, 97, 25 - 54.

Tian, K., et al. (2023). Economic exposure to regional value chain disruptions: evidence from Wuhan's lockdown in China. Regional Studies 57, 525-536.

-The results of the paper suggest a minor impact of MNEs on the Gini coefficient, with changes as minimal as from 0.22 to 0.23 in 2002, or from 0.23 to 0.25 to 2007. It is crucial to determine whether these changes are statistically significant. The manuscript would benefit from statistical tests to substantiate the claimed impact of MNEs on changes in carbon inequality from 2002-2012 and a decline in 2017.

Response: *Thank you for the suggestion. We have added a DM (Diebold-Mariano) test to check whether the original and hypothetical C-Ginis are statistically significantly different. We calculate both original and hypothetical C-Ginis for China as a whole and for its four zones (Eastern, Central, Western, and Northeastern China) for five years (1997, 2002, 2007, 2012, and 2017). The results are provided in Supplementary Table S5. We use this sample to conduct the DM test and the results confirm that they are statistically significantly different.*

Furthermore, as we explained above and will demonstrate again below, we have carefully verified the validity of our method. Therefore, we believe that the result is accurate.

-The manuscript currently calculates the Gini coefficient using a single indicator (e.g., 0.22, 0.23, 0.25), which does not differentiate the contributions from different components, such as domestic enterprises, Hong Kong, Macau, and Taiwan enterprises, or foreign enterprises. This aggregation limits the ability to substantiate claims regarding the impact of MNEs on inequality thoroughly. It would be beneficial to employ a decomposition method to quantify the individual contributions of each type of enterprise to the Gini coefficient, thus providing a more nuanced understanding of the sources of inequality.

Response: As the response to the last comment, we have calculated the contributions of the two types of MNEs (Hong Kong, Macau, and Taiwan enterprises, and foreign-owned enterprises), separately. We also calculate both original and hypothetical C-Ginis for China's four zones (Supplementary Table S5). We further examine which industries contribute most to the difference between the original and hypothetical inequality. We have added more nuanced analysis following the calculation. We analyze the effects of MNEs on carbon inequality by examining several key factors, including the geographical distribution of MNEs, the types of industries in which they operate, the differences in emission intensity between MNEs and domestic-owned enterprises, and regional disparities in emission intensity.

Please refer to pages 11-14 for our revision.

-The specific role of MNEs in carbon inequality levels seems the central theme of the article. Thus, it would be better to use some quantitative analysis to support the narrative of MNEs' role in inequality in China. Including more detailed charts and tables would strengthen this part of the study and provide clearer evidence.

Response: Our article is structured around three key components, each of which explores a different aspect of MNEs' role in inequality in China. We have provided detailed figures and tables to demonstrate our results. More tables are provided in the Supplementary information.

First, we account for the unequal contributions of MNEs to regional value-added and carbon emissions, thereby establishing a basis for understanding their role in carbon inequality (Figure 1, 2, and 3, and Supplementary Table S3).

Second, we analyze the impact of MNEs on carbon inequality using the C-Gini coefficient. We conduct detailed quantitative analyses by calculating both original and hypothetical C-Ginis for China as a whole (Figure 4), as well as its four zones for the five years (Supplementary Table S5). We also identify the industries that most significantly contribute to the difference between the original and hypothetical inequality.

Third, we conduct comprehensive econometric analyses regarding the effects of MNEs on the emission coefficients from a dynamic perspective (Table 1 and Supplementary Table S6). The regression results indicate that MNEs help narrow the emission coefficient gap between domestic-owned enterprises and MNEs. Following this, we conduct various scenario analyses to explore the potential of carbon emission reduction and inequality mitigation by leveraging the spillover effects of MNEs (Figure 5).

2. Method. The manuscript employs regression analysis to examine the role of MNEs in reducing carbon emission intensity. There are two set of dependent variables: (1) industry emission intensity; (2) the ratio of domestic firm emission intensity to MNE emission intensity. The independent variable considered is the proportion of MNE intermediate inputs in an industry. The logic here is that: (1) if industries utilizing more MNE inputs demonstrate lower emission intensities, it suggests that MNEs can help reduce emissions; (2) if there are narrower gaps in emission intensity between domestic and foreign firms when more MNE intermediate inputs are used, it implies MNEs can

lessen emission intensity gaps. However, this analysis might have conflated correlation with causality, and the relationship observed could be influenced by unaccounted variables, such as industry-specific technologies or regulatory environments.

Response: *Thanks for pointing out the possible unaccounted variables. To alleviate this concern, we have added a set of control variables in our regression. Considering scale effects, sectoral composition effects, and technique effects are the main channels through which a region's emission performance is determined, we introduce some control variables including production scale, industrial structure, technology innovation level, and environmental regulation level. We also include province, industry, and time fixed effects in the regressions.*

Please refer to pages 14-17 in the revised main text for our revision. We also provide a detailed introduction to our regression in Supplementary information 2.

-The current findings in the manuscript could indeed be influenced by endogeneity and reverse causality, which complicate the interpretation of the effects of MNEs on emission intensity. For instance, industries with inherently lower emission intensities might naturally attract more MNEs due to factors such as alignment with global environmental standards or the industry's higher technology and service-oriented focus. These industries, like service sectors or high-tech industries, typically have lower emissions compared to more traditional, labor-intensive industries. Additionally, in such sectors, the gap in emission intensity between domestic and foreign firms might already be minimal due to the nature of the industry itself, rather than the influence of MNEs. Therefore, without addressing these potential endogeneity issues through robust statistical methods like instrumental variables or fixed effects models, the regression results might overstate the impact of MNE inputs on reducing emission disparities.

Response: *We address potential endogeneity and reverse causality by using lagged explanatory variables and an instrumental variable: province-level Foreign Market Access. This variable is measured by the distance from a province's capital city to its nearest port. The rationale for the variable is that provinces closer to a port are more likely to have better access to foreign markets, making them more attractive to MNEs. Additionally, a region's emission efficiency is not related to its geographical characteristics, making this an appropriate instrumental variable. We also check the robustness of the results by employing an alternative estimator-GMM to partially solve the endogeneity problem.*

Please refer to pages 14-17 in the main text for our revision.

-The manuscript discusses two potential mechanisms by which MNEs could influence a country's emission intensities. However, the current regression analysis does not adequately determine which mechanism dominates. The two impacts mentioned as 'the pollution haven effect'—increasing the emission coefficient—and 'the knowledge spillover effect'—leading to technological upgrading that reduces the emission coefficient cannot be justified or ruled out based on the above regression. For instance, if MNEs contribute to a "pollution haven" effect, where domestic firms increase

emissions in sectors that are more upstream (which may use fewer MNE intermediates), the findings could still align with the observed results: those who use less MNEs inputs can have a higher emission intensity. However, this is not induced by the ‘knowledge spillover effect’. Instead, it is due to the contrary effect. Thus, the current regression setup cannot provide enough evidence to distinguish between these mechanisms.

Response: *The major objective of the regression analysis is to demonstrate that MNEs narrow the emission coefficient gap between domestic-owned enterprises and MNEs so as to provide evidence for the subsequent scenario analysis. Upon confirming that MNEs contribute to the convergence of domestic-owned enterprises’ emission efficiency with that of MNEs, we conduct different scenario analyses to explore the potential of carbon emission reduction and inequality mitigation by leveraging MNEs. Testing the pollution haven effect is not our primary objective and lies beyond the scope of this paper. Instead, we concentrate this paper on the effects of MNEs on the emission coefficient and the coefficient gap. We believe that the knowledge spillover effect is the most likely mechanism through which MNEs facilitate the convergence of emission efficiency.*

We have accordingly rewritten the sentences. Please refer to pages 14-17 in the main text for our revision.

-In assessing the robustness of the regression analysis in your manuscript, it's essential to consider whether robust error terms are being used. If robust errors are not being applied, possible violations of regression assumptions such as heteroscedasticity or autocorrelation may bias standard error estimates and influence the significance levels of coefficients. Clustering standard errors at appropriate levels, such as industry or regional levels, can further enhance the analysis. This particularly important when the data is across different industries or geographical areas, to control for unobserved heterogeneity within these groups.

-Temporal and Spatial Autocorrelation.

-Please report the number of observations as well.

Response: *Thanks for the comments above. In our revised manuscript, we validate the robustness of the results through the following approaches: (i) comparing outcomes without and with control variables (column 1 and 2); (ii) employing robust standard errors clustered at the regional (column 3) and industry (column 4) levels; (iii) utilizing instrumental variables (column 5); (iv) applying an alternative estimation method—the GMM (column 6). Potential temporal and spatial autocorrelation have been considered and the number of observations is reported in all the regressions.*

Please refer to pages 14-17 in the main text for our revision.

3. Method. In the scenario analysis, the current method approach involves directly reducing the emission intensity of the highest emitters. For instance, scenario 1 directly assumes that the domestic enterprises in the western regions are to have the same emission intensities of MNEs. This can lead to a straightforward result—lower inequality levels, as it just flattens the Lorenz curve.

-However, considering an alternative approach to "leveraging MNEs" that involves further lowering the emission intensities of provinces on the left of the Lorenz curve could present different outcomes. For example, if the scenario assumes that domestic enterprises in the eastern regions have the same emission intensities of their own MNEs, it might actually increase inequality.

-Why the certain scenarios in the manuscript are chosen instead of the alternative proposed? It would be better to have some justifications to support the scenario settings.

***Response:** Thanks for the questions above. The rationale behind the scenarios setting is as follows. We first conduct regression analysis and reveal that the presence of MNEs facilitates the convergence of domestic-owned enterprises' emission coefficient with that of MNEs. In other words, MNEs enable technological laggards to catch up with industry leaders. The emission coefficient gaps between domestic-owned enterprises and MNEs are more pronounced in Western China, Central China, and Northeastern China than those in Eastern China. Therefore, it is more feasible and urgent for domestic enterprises in these inland regions to converge to the emission technology level of MNEs. This underpins our scenarios setting: domestic-owned enterprises in carbon-intensive industries in Western China converge to those of MNEs within the same industry and province; the emission coefficients of domestic-owned enterprises across all industries in Western China converge to those of MNEs; the emission coefficients of domestic-owned enterprises in carbon-intensive industries in Western China, Central China, and Northeastern China converge to those of MNEs. These scenarios aim to investigate potential policy implications for leveraging MNEs, particularly in inland regions where convergence is more practical and feasible, to maximize emission reductions and inequality mitigation.*

Domestic enterprises in eastern regions may also potentially converge to the emission intensities of local MNEs. However, this scenario offers lower marginal potential for emission reduction and inequality mitigation (even increasing inequality) because domestic enterprises in eastern regions already possess cleaner technologies compared to those in inland regions. Therefore, we propose that policy initiatives could prioritize upgrading emission technologies in inland regions by attracting MNEs to increase investments in these areas.

In our revised manuscript, we added an additional scenario where domestic enterprises in inland regions converge to the emission technology levels of those in eastern regions. This scenario considers the possibility that MNEs may not increase their investment in inland regions as expected (due to economic and geopolitical issues), while domestic enterprises in eastern regions may expedite the transfer of advanced emission technologies towards inland regions.

We have added more justifications regarding our scenario settings. Please refer to pages 18-19 for our revision.

4. Method. The hypothetical extraction method (HEM) is chosen instead of the decomposition method.

-Why HEM is chosen over the decomposition method? Are there any particular advantages of choosing HEM over decomposition?

-Although it is stated here that the HEM and decomposition methods would yield consistent results, are there any mathematical proof for this?

Response: There are two widely used methods to calculate the contribution of MNEs: the decomposition method (Koopman et al., 2014) and the hypothetical extraction method (HEM, Los et al., 2016; Dietzenbacher et al., 2019). The two methods yield the same results. We provide the mathematical proof for it in Supplementary information 3. However, Los et al. (2016) note that the decomposition method is overly complex in terms of both mathematical derivation and interpretation and propose that the more intuitive HEM can be more easily applied to the data. Therefore, we present the HEM in the main text.

Koopman, R., Wang, Z., & Wei, S.J. (2014). Tracing value-added and double counting in gross exports. *American Economic Review*, 104(2), 459-494.

Los, B., Timmer, M. P., & de Vries, G. J. (2016). Tracing value-added and double counting in gross exports: Comment. *American Economic Review*, 106(7), 1958-1966.

Dietzenbacher, E., van Burken, B. & Kondo, Y. (2019). Hypothetical extractions from a global perspective. *Economic Systems Research* 31, 505-519.

-How does this study deal with this? Under the HEM, emissions are regarded as related to domestic firms/HMT firms/foreign firms. However, if I understand correctly, there are some parts of emissions that would be quantified as related to HMT firms and as related to foreign firms as well. For instance, considering the simplest case that there is only 1 region, 1 sector, and 2 types of firms.

$$A = \begin{bmatrix} \text{Add} & \text{Adf} & \text{Afd} & \text{Aff} \end{bmatrix}$$

$$A' = \begin{bmatrix} \text{Add} & 0 & 0 & 0 \end{bmatrix}$$

$$A'' = \begin{bmatrix} 0 & 0 & 0 & \text{Aff} \end{bmatrix}$$

$$F = \begin{bmatrix} \text{Fdd} & \text{Fdf} & \text{Ffd} & \text{Fff} \end{bmatrix}$$

$$F' = \begin{bmatrix} \text{Fdd} & 0 & 0 & 0 \end{bmatrix}$$

$$F'' = \begin{bmatrix} 0 & 0 & 0 & \text{Fff} \end{bmatrix}$$

So, if the total output is represented by AF, and the portion unrelated to foreign investment is A'F', will subtracting A'F' from AF result in the foreign-influenced part? If the portion unrelated to domestic investment is A''F'', will subtracting A''F'' from AF result in the domestic-related part? Do these two components added together still equal the total output?

Response: Thank you for the question. We calculate the contributions of MNEs by simultaneously extracting HMT and foreign firms. The respective contributions of HMT firms and of foreign firms are calculated by extracting HMT firms and foreign firms separately. It is true that the summation of HMT contributions and foreign firm contributions is slightly larger than MNEs contributions. This is because there are industrial linkages between HMT firms and foreign firms. When calculating the contributions of HMT firms, this part of linkages is extracted. It is extracted again when calculating the contributions of foreign firms. Therefore, adding up the contributions of HMT firms and of foreign firms causes double counting. This is a common issue in similar input-output accounting work. A typical example is the work of Koopman et al. (2014). They decompose the gross exports into 16 items, one of which is double counting. Generally, the double counting is very limited. We also checked in our case that the double counting is less than 1%. To avoid the limited double counting, when calculating the contributions of MNEs, we simultaneously extracting HMT and foreign firms instead of adding up their respective contributions.

Koopman, R., Wang, Z., & Wei, S.J. (2014). Tracing value-added and double counting in gross exports. *American Economic Review*, 104(2), 459-494.

5. Results. Figure 4 states that ‘China has undergone an escalating regional carbon inequality during 2002–2017, as evidenced by the rise in its original C-Gini coefficient from 0.23 in 2002, to 0.25 in 2007, 0.28 in 2012, and 0.32 in 2017.’

-Is the C-Gini based on consumption-based emissions, production-based emissions, or the HEM method emissions?

Response: *Both the original C-Gini and the hypothetical C-Gini are calculated based on production-based emissions. We have made it clear in the methods section.*

-This study finds an increase in carbon inequality. How does this compare with other similar studies on regional carbon inequalities? Can this finding be reconciled with previous studies that observed a decrease in inequality? How can the differences be explained, e.g., are they due to the differences between the residential sector and overall final demand, or are there any other factors?

Response: *The results of carbon Gini coefficients in our paper are consistent with those from previous studies. For example, Zhang et al. (2023) also report that China’s national carbon inequality increased during 2007 – 2017. Meanwhile, it is worth noting that the uncertainties in the Gini results may stem from the CO₂ emission inventory. Our emission inventory is sourced from the CEADs database. According to the database developers (Shan et al., 2018; Shan et al., 2020), there are several factors that can introduce different levels of uncertainty into the emission data including the activity data, emission factors, lack of completeness, and measurement errors. They use Monte Carlo simulation to assess the uncertainty in the emission data and update the emission data when new information becomes available. Despite these uncertainties, their estimates are considered reliable and are widely used by researchers.*

We have added some discussion regarding the uncertainties in the Supplementary information 1. Please refer to the new manuscript for our revision.

Zhang, H., et al. (2023). Worsening Carbon Inequality Embodied in Trade within China. *Environ. Sci. Technol.* 57, 863 – 873.

Shan, Y. et al. China CO₂ emission accounts 1997 – 2015. *Scientific data* 5, 1-14 (2018).

Shan, Y., Huang, Q., Guan, D., Hubacek, K. China CO₂ emission accounts 2016 – 2017. *Scientific data* 7, 54 (2020).

Minor

-As China's entry into the WTO happened in 2001, can the period from 2002 to 2017 fully capture the rapid expansion of foreign investment and MNEs? Given that an inter-provincial IO table for China has already been established covering the long period from 1987 to 2017, it might be worthwhile to extend the analysis to include more years. Including data from 1997 as a comparative benchmark might provide additional insights.

Response: *We have included data from 1997 in our revised manuscript. All related results have been revised accordingly.*

-Page 5 line 125, Figure 1. It would be better if both Figure 1 a and b use the same color legend to represent the same item for clarity and consistency.

Response: *We have revised the figures as suggested.*

-Page 7 line 167-170. The statement that 'from 2002 to 2017, there was a distinct shift in the geographical distribution of value-added generated by MNEs, marking a transition from the east to the west' might be reconsidered. The change from 71.2% to 67.6% over fifteen years could be perceived as relatively minor. It may be more accurate to describe this shift as gradual rather than 'distinct' to reflect the modest scale of change more precisely.

Response: *We have included data from 1997 and rewritten the sentences.*

-Page 11 line 249-250. The statement that 'The exacerbated carbon inequality over the 2002–2017 period is determined by changes in the cumulative proportions of CO₂ emissions and value-added' seems redundant. This is because by definition the Gini coefficient inherently measures inequality based on cumulative proportions, describing this as a novel finding may not be necessary. It would be more insightful to contextualize how these specific changes contribute uniquely to carbon inequality or to explore additional factors that might have influenced the trends observed during this period.

Response: *Thank you for pointing out the problem. We have revised the sentences.*

-Page 11, lines 260-265. Although line 249-250 shows that 'from 2002 to 2017, the eastern regions experienced a faster economic development', while the provided example here argues that Ningxia (a western region) experienced a 736.4% increase in value-added, compared to Guangdong's (an

eastern region) 564.2% increase. This seems to contradict the general statement about regional economic growth rates. To avoid confusion and strengthen the argument, it might be beneficial to select examples that align more closely with the initial claim about regional growth patterns, or to clarify why these particular examples were chosen despite the apparent discrepancy.

Response: *We have revised the sentences and selected another example.*

-P23 Equation 6. Since this manuscript specifically addresses carbon inequality, it would be more appropriate to replace the equation for the Gini coefficient of income inequality with one that directly can be applied to estimate Chinese carbon inequality.

Response: *We have revised the equation. Thank you for the suggestion.*

Reviewer #1 (Remarks on code availability):

No code shared at this stage.

Response: *We have shared the code.*

Reviewer #2 (Remarks to the Author):

This study focused on China and uncovered the relationships between its exacerbated carbon inequality and MNEs. While this kind of work is necessary, I believe that this manuscript does not meet the publication criteria of Nature Communications for the following reasons.

1. The research background section puts forward the scientific problem of this research more clearly, but the sorting out of existing research is weak, and the discussion of whether this research has important theoretical value or application prospects is insufficient.

Response: *Thanks for your comments. In the revised manuscript, we have carefully addressed your concerns from the following aspects.*

Firstly, we undertook comprehensive literature review. *Following the author guidelines of Nature Communications, we briefly summarized the relevant literature into three main strands. The first strand of literature focused on the inequality of economic benefits and environmental burdens embodied in international or intranational trade. The second strand extended the concentration from trade to FDI and has traced the carbon footprints of MNEs at both national and global levels. The third strand has estimated the inequality of carbon emissions across different income groups or age groups. However, few studies have explored the impacts of MNEs on economic gains, environmental burdens, and the resultant inequality at a sub-national level.*

Secondly, the contributions of this paper are further elucidated. *Specifically, our quantitative research makes three primary contributions. First, our study pioneers in investigating the environmental effects of MNEs at the provincial level, thereby enhancing our comprehension of how MNEs can affect China's balanced regional development in terms of both economic gains and environmental burdens. Second, we employ innovative methods, including a novel environmentally extended interprovincial input-output (IPIO) model that distinguishes China's enterprises in each province-industry pair into three ownership types. Third, our findings provide valuable insights into potential strategies for China to accomplish its dual control goals—reducing both emission intensity and total emission amount.*

Please refer to the responses to the question below for a more detailed introduction to the novelty and contributions of this paper.

Please refer to pages 3-4 for the revision.

2. The introduction does not explain whether there is any innovation in theory, method or technology in this research, and the innovation of the research is insufficient. What is the significance of quantitative research on the impact of MNEs on carbon inequality?

Response: *Thanks for your comments and questions. In the revised manuscript, the novelty and contributions of this paper are further strengthened.*

The concept of regional carbon inequality captures the disparities in emission intensity across different regions. Our quantitative research on the impact of MNEs on carbon inequality enhances our understanding of how MNEs can facilitate China's achievement of reducing emission intensity

and emission amount, thereby fostering balanced regional development.

Specifically, the novelty and contributions of this paper are threefold.

Firstly, as one of the pioneering studies to delve into the environmental effects of MNEs at the provincial level, this paper raises questions and findings that significantly differ from previous studies. This paper contributes to the existing literature by estimating how and to what extent MNEs in China contribute to its provincial value-added creation, carbon emissions, and the ensuing regional carbon inequality over the period of 1997-2017. This paper also investigates how MNEs may affect emission intensity and their potential to reduce emission intensity, emission amount, and carbon inequality. In the era of global value chains (GVCs), MNEs have been thrust to the forefront of both economic interaction and climate action due to their profound transnational impacts. Acknowledging the pivotal role of MNEs in climate action, several previous studies have traced the carbon footprints of MNEs at national and global levels (López et al., 2019; Zhang et al., 2020). However, our research is the first to explore the environmental effects of MNEs at the provincial level in China. Given the pronounced provincial heterogeneity and significant inter-provincial economic connections in China, it is imperative to incorporate provincial-level differences and inter-provincial connections when investigating the environmental effects of MNEs. The provincial-level analyses provide crucial insights for advancing China's green development. In pursuit of carbon neutrality, China has recently released a circular outlining a plan to establish a dual control system for carbon emissions, with a primary focus on emission intensity, supplemented by the control over total emission amount. According to the circular, China is poised to enhance its emission calculation and carbon footprint accounting for regions (provinces and cities), sectors and key enterprises. Responsibilities for dual control will be reasonably allocated to each region, key industries and enterprises. In this context, **our provincial-level study contributes to a more comprehensive understanding of how MNEs may facilitate the achievement of dual control goals.**

Secondly, we propose to address the questions using multiple quantitative models including a novel environmentally extended interprovincial input-output (IPIO) model that distinguishes China's enterprises in each province-industry pair into three types by ownership. We employ the novel IPIO model to account for the contributions of MNEs to the value-added creation and carbon emissions in each China's province. This provides a scientific methodological framework to account for the carbon footprints for regions (provinces and cities), sectors and key enterprises (in our case mainland China-, Hong Kong, Macao, and Taiwan-, and foreign-owned enterprises, respectively). We propose to use the hypothetical extraction method to assess the impact of MNEs on carbon inequality. The identical method and analytical framework can be extended to analyze relevant issues on a global scale and be applied to other countries facing notable regional disparities. We further use panel data regression model to identify the effects of MNEs on emission intensity. Following the finding of the regression that MNEs facilitate the convergence of domestic-owned enterprises' emission coefficient with that of MNEs, we conduct scenario analyses to explore the potential of leveraging MNEs to reduce emission intensity, emission amount, and carbon inequality.

Thirdly, the findings provide insights into potential strategies that China could adopt to facilitate the achievement of its dual control goals—reducing both emission intensity and total emission amount. First, our accounting results indicate that MNEs significantly contribute to China's CO₂, thereby necessitating them to assume their carbon mitigation responsibilities. We emphasize that establishing a sound carbon emission accounting system is a prerequisite for assigning the mitigation responsibilities to different regions, key industries and enterprises, thereby

supporting China's dual control efforts. **Second**, we find that the increasing presence of MNEs in inland regions has contributed to balanced economic development and mitigated regional carbon inequality. Therefore, we emphasize the need for intensified efforts to attract more foreign investment to Central, Western, and Northeastern China. **Third**, our scenario analyses underscore the imperative to enhance low-carbon technologies in inland regions, particularly within carbon-intensive industries. MNEs should share environmental responsibility in their host regions and strengthening coordination between MNEs and local enterprises is imperative to expedite the diffusion of cleaner production technology. If MNEs reduce their investments in China or its inland regions, disrupting the technology spillover effects to local enterprises, it is essential to encourage coastal domestic-owned enterprises to increase their investments in clean technologies inland so as to accelerate the transfer of knowledge and advanced emission-reducing technologies to these regions.

We have highlighted the novelty and contribution in the Introduction on page 4. Please refer to pages 21-22 for our discussion regarding policy implications.

López, L.-A., Cadarso, M.-Á., Zafrilla, J. & Arce, G. (2019) The carbon footprint of the US multinationals' foreign affiliates. *Nature communications* 10, 1672.

Zhang, Z. et al. (2020) Embodied carbon emissions in the supply chains of multinational enterprises. *Nature Climate Change* 10, 1096-1101.

3. Does the innovation of this paper only increase the impact of MNEs on carbon inequality compared to the following studies?

Zhang, W., Liu, Y., Feng, K., Hubacek, K., Wang, J., Liu, M., Jiang, L., Jiang, H., Liu, N., Zhang, P., Zhou, Y., Bi, J., 2018a. Revealing Environmental Inequality Hidden in China's Inter-regional Trade. *Environ. Sci. Technol.* 52, 7171–7181.

Zhang, H., Zhang, W., Lu, Y., Wang, Y., Shan, Y., Ping, L., Li, H., Lee, L.-C., Wang, T., Liang, C., Jiang, H., Cao, D., 2023. Worsening Carbon Inequality Embodied in Trade within China. *Environ. Sci. Technol.* 57, 863–873.

Response: Thank you for the comment. We have carefully read the papers and compared them with our paper. As we have discussed in the response to Question 2, our study differs significantly in at least three aspects.

First, our research focus is different. Both Zhang et al. (2018) and Zhang et al. (2023) focused on inter-regional trade within China. Instead, we estimate how and to what extent MNEs (FDI) in China contribute to its provincial value-added creation, carbon emissions, and the ensuing regional carbon inequality over the period of 1997 – 2017. We also examine how MNEs may affect emission intensity and explore their potential to reduce emission intensity, emission amount, and carbon inequality. **Second, our model is different.** In this paper we adopt both input-output (IO) model and econometric model. The IO model is a novel environmentally extended IPIO that distinguishes China's enterprises in each province-industry pair into three types by ownership. **Third, our findings and implications are different.** As delineated above, our findings provide insights in at least three aspects into potential strategies that China could adopt to facilitate the achievement of

its dual control goals.

Therefore, our study does not overlap with these papers, and it has its own novelty and contribution in terms of research focus, models, findings, and implications.

4. The research methodology of this paper is more reasonable and can basically solve the scientific problem of this paper. However, there is no mention of the advantages of using the Gini coefficient to characterize carbon inequality, nor is interregional carbon inequality taken into account.

Response: Thank you for the comment. There are several advantages of using the Gini coefficient to characterize carbon inequality.

First, the Gini coefficient provides intuitive features that help make understand inequality more easily as it ranges from 0 to 1. The Gini coefficient is a measure of statistical dispersion with a range from 0 to 1, where 0 represents perfect equality and 1 represents perfect inequality. As a result, it is best for providing a quick, intuitive measure of inequality. We are aware of several other measures of inequality such as the Theil index. The Theil index is an entropy-based measure of inequality and it ranges from 0 to infinity, where 0 indicates perfect equality. Higher values indicate more inequality, but there is no upper bound. The values of the Theil index are less intuitive than those of the Gini coefficient because there is no fixed upper limit.

Second, the Gini coefficient is widely used in research on inequality so that we can compare our results with those in previous studies. For example, Zhang et al. (2023) also use the Gini coefficient to measure carbon inequality and report that China's carbon inequality increased during 2007 – 2017. Our results are consistent with these previous studies.

With regards to the interregional carbon inequality, it is worth noting that the inequality caused by interregional linkages is taken into account as we employ an environmentally extended interprovincial input-output (IPIO) model to calculate the Gini coefficient. IPIO model enables us to capture both direct and indirect interregional effects. In the Results section, we also analyze the interregional carbon inequality among the four zones (Eastern, Central, Western, and Northeastern China).

Zhang, H., Zhang, W., Lu, Y., Wang, Y., Shan, Y., Ping, L., Li, H., Lee, L.-C., Wang, T., Liang, C., Jiang, H., Cao, D., 2023. Worsening Carbon Inequality Embodied in Trade within China. *Environ. Sci. Technol.* 57, 863 – 873.

5. In measuring carbon inequality (C - Gini), this paper replace the income with carbon emissions and the population with value-added, please explain its rationality.

Response: Thank you for the comment. The rationale behind is the fact that our calculation goal coincides with the connotation of the Gini coefficient. Specifically, the income Gini coefficient measures the inequality among different income levels, and it is built by comparing the cumulative proportions of the population against the cumulative proportions of income they receive. If a small

proportion of the population possesses all the income while the rest has none, it results in extreme inequality. In this paper, we aim to assess the inequality in CO₂ emissions across China's regions relative to their value-added gains. If regions with minimal value-added emitted the majority of the CO₂ emissions, this would also represent extreme inequality. Therefore, we build the C-Gini by comparing the cumulative proportions of value-added against the cumulative proportions of CO₂ the regions emit. Similarly, some previous studies such as Zhang et al. (2023) also employed the C-Gini coefficient to measure the emission inequality relative to value-added.

We have explained the rationality clearly and revised the sentences. Please refer to page 25-26 for our revision.

Zhang, H., Zhang, W., Lu, Y., Wang, Y., Shan, Y., Ping, L., Li, H., Lee, L.-C., Wang, T., Liang, C., Jiang, H., Cao, D., 2023. Worsening Carbon Inequality Embodied in Trade within China. *Environ. Sci. Technol.* 57, 863 – 873.

6.The resolution of Figure 1 is low, and it is not clear and easy to read.

Response: Thank you. We have improved the figures and the high-resolution figures are also provided as separate files.

7.It is suggested to abbreviate the names of provinces in Fig3-Fig5.

Response: Thanks for the suggestion. To enhance the readability of the figure, we chose to retain the full names of the provinces. This is due to the fact that many readers may not be familiar with Chinese province, thereby using abbreviations could potentially cause confusion. Nevertheless, we made adjustments to the figures so that it does not look crowded with full names of provinces. Similar naming conventions have also been adopted by previous studies (Zhang et al., 2018; 2023).

Zhang, W., et al. (2018). Revealing Environmental Inequality Hidden in China's Inter-regional Trade. *Environ. Sci. Technol.* 52, 7171 – 7181.

Zhang, H., et al. (2023). Worsening Carbon Inequality Embodied in Trade within China. *Environ. Sci. Technol.* 57, 863 – 873.

8.It is mentioned in the part of result analysis that reducing carbon emission coefficient can reduce carbon inequality, but reducing carbon emission coefficient can only reduce carbon emission to a certain extent. How to further reduce carbon inequality needs to be analyzed in combination with the transfer of value-added.

Response: Thanks for your comments. We agree that reducing carbon inequality involves both the changes in carbon emissions and value-added. In this paper, we conduct scenario analyses by

adjusting emission coefficients, in particular, by narrowing the emission coefficient gaps between inland regions and coastal regions. A decrease in the emission coefficient is widely regarded as an advancement in emission technology. Therefore, it is feasible to adjust the emission coefficient by enhancing the emission technology. For example, if a company located in inland regions adopts more advanced emission technology from its multinational counterparts or a similar company in a coastal region, it can improve its technology and lower its emission coefficient. Before the scenario analyses, we also employ panel data regression models to provide evidence that the presence of MNEs facilitate the technology improvement of local enterprises. In this way, we provide policy implications regarding leveraging MNEs to improve emission technology, thereby reducing emission amount and inequality. Indeed, changes in emission technology may also affect the creation of value-added. However, this is beyond the scope of our study.

9. There are many factors affecting carbon inequality. On the whole, the research content of this paper is insufficient and lacks depth. It is suggested to explore the deep causes of carbon inequality changes.

Response: *Thank you for the suggestion. In the revised manuscript, we have significantly enhanced the elaboration on the role of MNEs in carbon inequality from multiple dimensions.*

The source of MNEs' impacts on carbon inequality is that MNEs generate uneven value-added and carbon emissions across China's regions. Therefore, we systematically organize a coherent sequence as follows to provide a nuanced interpretation of the role of MNEs in carbon inequality.

We first account for the unequal contributions of MNEs to regional value-added and carbon emissions, establishing a basis for understanding their role in carbon inequality. We analyze the effects of MNEs on carbon inequality by examining several key factors, including the geographical distribution of MNEs, the types of industries in which they operate, the differences in emission intensity between MNEs and domestic-owned enterprises, and regional disparities in emission intensity. We also assess the contributions of the two types of MNEs (Hong Kong, Macau, and Taiwan enterprises, and foreign-owned enterprises), separately, and compare the original and hypothetical C-Gini for China's four zones (Supplementary Table S5). We further examine what industries contribute mostly to the difference between the original and hypothetical inequality. We have added more nuanced analysis following the calculation.

Please refer to the new manuscript for our revision.

10. The policy recommendations for reducing carbon inequality in the discussion appeared to be too general.

Response: *Thank you for the comment. In the revised manuscript, we have carefully enhanced and refined the content of policy recommendations.*

Our policy recommendations are primarily derived from our empirical findings and scenario analyses. For example, we find that increasing presence of MNEs in inland regions has contributed

to balanced economic development and mitigated regional carbon inequality. Therefore, we emphasize the need for intensified efforts to attract more foreign investment to inland regions.

In our revised manuscript, we consider the possibility that MNEs may not increase their investment in inland regions as expected due to economic and geopolitical issues, thereby disrupting the technology spillover effects on local enterprises. Therefore, we added one additional scenario where domestic enterprises in inland regions converge to the emission technology levels of those in eastern regions. The results demonstrate a substantial decline in carbon emissions and an impressive reduction of in the C-Gini. We thus stress the importance of stimulating domestic-owned enterprises in coastal regions to increase their investments in clean technologies inland if foreign investments would be disrupted.

Please refer to pages 20-22 for our detailed revision.

11. What are the limitations of this study and future work? They should be supplemented.

Response: Thank you for the suggestion. The limitations of this study and prospects for future work have been added in the revised manuscript.

The limitations of this study may stem from the uncertainties of the province-industry CO₂ emission inventory. Our emission inventory is sourced from the CEADs database. According to the database developers (Shan et al., 2018; Shan et al., 2020), factors such as the activity data, emission factors, lack of completeness, and measurement errors can introduce different levels of uncertainty. They use Monte Carlo simulation to assess the uncertainty in the emission data and update the emission data when new information becomes available. Despite these uncertainties, their estimates are considered reliable and are widely used by researchers.

With regard to extension for future work, the scheme and methodology in this study can be applied to more countries and extrapolated to a global scale. For countries with substantial regional disparities, our methodology can be utilized to investigate their regional carbon inequality.

We have supplemented the extension for future work in the last paragraph, and clarification of data uncertainties in the data section (Supplementary information 1).

Shan, Y. et al. China CO₂ emission accounts 1997 – 2015. *Scientific data* 5, 1-14 (2018).

Shan, Y., Huang, Q., Guan, D., Hubacek, K. China CO₂ emission accounts 2016 – 2017. *Scientific data* 7, 54 (2020).

12. Note that references should be formatted consistently.

Response: We have formatted the references in a consistent way. Thank you again for your comments.

Response to the Reviewers

From: “Leveraging multinational enterprises to reduce the escalating regional carbon inequality in China” (NCOMMS-24-21432A)

The authors are grateful for the insightful comments and constructive suggestions from the reviewers. We have carefully revised the manuscript accordingly, which we believe has made for a better article. Please find below our detailed point-by-point responses to the reviewers. To enhance readability, we have copied the feedbacks of the reviewers and put our responses in italics.

REVIEWER COMMENTS

Reviewer #1 (Remarks to the Author):

After reading the paper and the authors' reply thoroughly, I am satisfied that they have addressed the issues raised in previous version. My recommendation is therefore for the paper to be accepted for publication with minor revisions.

-Figure 2. The numbers in the circles are difficult to read. Please either increase the font size or use a different color.

Response: Thank you for the suggestion. We have increased the font size and also changed the color to enhance the readability of the numbers in the circles.

-Figure 3. In “value-added/CO₂ emissions”, please check whether “and” should be used. Also, please delete extra “ ” in the caption.

Response: Thank you for the suggestion. We have modified “value-added/CO₂ emissions” to “value-added or CO₂ emissions”. We have also removed all extra blank spaces in the caption.

-Figures 4 and 5, remove the extra “ ” in the caption.

Response: We have removed all extra blank spaces in the caption. Thank you.

-Please move the DM (Diebold-Mariano) test section from the supplementary file to the method section.

Response: Thank you for the suggestion. We have added a paragraph introducing the DM test in the method section.

-Please add a section about the limitations in the method section, which have already been discussed in your response letter.

Response: Thank you for the suggestion. We have added a section discussing the limitations in the last part of the method section.

Reviewer #2 (Remarks to the Author):

Thanks to the author's response, my questions have been answered. However, I still believe that this manuscript does not meet the publication criteria of Nature Communications for the following reasons:

1. Although the authors emphasize the novelty and contributions of this paper, I still believe that on the basis of existing research, this article does not represent a significant breakthrough.

Response: Thank you for the comment. Our article addresses the critical issue of inequality and climate change (critical SDGs) by exploring a potential solution: leveraging the spillover effects of multinational enterprises (MNEs) to reduce the escalating regional carbon inequality. We tackle this challenge employing novel quantitative models. Given the significance of our research topic, the novelty of our methodology, and the inspiring findings, we believe it is well-suited for publication in Nature Communications.

The novelty and contributions of this paper can be specifically outlined in the following three aspects, all of which have also been thoroughly elaborated in the manuscript.

Firstly, our paper addresses a critical issue in sustainable development, specifically the role of MNEs in mitigating inter-provincial inequality and combating climate change. Our research is the first to explore both the economic and environmental effects of MNEs at the provincial level in China. This paper contributes to the existing literature by estimating how and to what extent MNEs in China contribute to its provincial value-added creation, carbon emissions, and the ensuing regional carbon inequality over the period of 1997-2017. Moreover, this paper also investigates how MNEs may affect emission intensity and their potential to reduce emission intensity, emission amount, and carbon inequality. Given the pronounced provincial heterogeneity and significant inter-provincial economic connections in China, it is imperative to incorporate provincial-level differences and inter-provincial connections when investigating the environmental effects of MNEs. The provincial-level analyses provide crucial insights for advancing China's green development. In pursuit of carbon neutrality, China has recently released a circular outlining a plan to establish a dual control system for carbon emissions, with a primary focus on emission intensity, supplemented by the control over total emission amount. According to the circular, China is poised to enhance its emission calculation and carbon footprint accounting for regions (provinces and cities), sectors and key enterprises. Responsibilities for dual control will be reasonably allocated to each region, key industries and enterprises. In this context, our provincial-level study contributes to a more comprehensive understanding of how MNEs may facilitate the achievement of dual control goals.

Secondly, we employ innovative quantitative models, including a novel environmentally extended interprovincial input-output (IPIO) model that distinguishes China's enterprises in each province-industry pair into three types by

ownership. We employ the novel IPIO model to account for the contributions of MNEs to the value-added creation and carbon emissions in each China's province. This provides a scientific methodological framework to account for the carbon footprints for regions (provinces and cities), sectors and key enterprises (in our case mainland China-, Hong Kong, Macao, and Taiwan-, and foreign-owned enterprises, respectively). We propose to use the hypothetical extraction method to assess the impact of MNEs on carbon inequality. The identical method and analytical framework can be extended to analyze relevant issues on a global scale and be applied to other countries with notable regional disparities. We further use a panel data regression model to identify the effects of MNEs on emission intensity. Following the finding of the regression that MNEs facilitate the convergence of domestic-owned enterprises' emission coefficients with those of MNEs, we conduct scenario analyses to explore the potential of leveraging MNEs to reduce emission intensity, emission amount, and carbon inequality.

Thirdly, the findings provide insights into potential strategies that China could adopt to facilitate the achievement of its dual control goals—reducing both emission intensity and total emission amount. **First,** our accounting results indicate that MNEs significantly contribute to China's CO₂, thereby necessitating them to assume their carbon mitigation responsibilities. We emphasize that establishing a sound carbon emission accounting system is a prerequisite for assigning the mitigation responsibilities to different regions, key industries and enterprises, thereby supporting China's dual control efforts. **Second,** we find that the increasing presence of MNEs in inland regions has contributed to balanced economic development and mitigated regional carbon inequality. Therefore, we emphasize the need for intensified efforts to attract more foreign investment to Central, Western, and Northeastern China. **Third,** our scenario analyses underscore the imperative to enhance low-carbon technologies in inland regions, particularly within carbon-intensive industries. MNEs should share environmental responsibility in their host regions and strengthening coordination between MNEs and local enterprises is imperative to expedite the diffusion of cleaner production technology. If MNEs reduce their investments in China or its inland regions, disrupting the technology spillover effects to local enterprises, it is essential to encourage coastal domestic-owned enterprises to increase their investments in clean technologies inland so as to accelerate the transfer of knowledge and advanced emission-reducing technologies to these regions.

2.The depth and breadth of this study is insufficient and does not significantly expand or deepen existing research.

Response: Thank you for the comment. Our study expands and deepens existing research in at least three key aspects.

Firstly, this paper is among the pioneering studies to delve into both the economic and environmental effects of MNEs at the provincial level, and our research topic goes beyond previous studies with a focus on the effects of MNEs on

regional carbon inequality. Acknowledging the pivotal role of MNEs in climate action, several previous studies have traced the carbon footprints of MNEs at national and global levels (López et al., 2019; Zhang et al., 2020). However, our research is the first to examine both the economic and environmental effects of MNEs at the provincial level in China. Moreover, while previous studies have primarily focused on tracing the carbon footprints of MNEs, we delve deeper into their effects on regional carbon inequality and raise findings that significantly go beyond previous studies. We analyze the effects of MNEs on carbon inequality by examining several key factors, including the geographical distribution of MNEs, the types of industries in which they operate, the differences in emission intensity between MNEs and domestic-owned enterprises, and regional disparities in emission intensity. We also assess the contributions of the two types of MNEs (Hong Kong, Macau, and Taiwan enterprises, and foreign-owned enterprises), separately. We further examine which industries contribute most to the difference between the original and hypothetical inequality.

Secondly, this paper is among the first to employ the novel environmentally extended IPIO model that distinguishes China's enterprises by ownership. Moreover, we conduct solid panel regression analyses to investigate the impact of MNEs in China on the emission coefficient, thereby providing empirical evidence for leveraging the spillover effects of MNEs to improve emission technology. We include a set of control variables, province, industry, and time fixed effects in our regression to alleviate the effects of potential omitted variables. We also address potential endogeneity and reverse causality by using lagged explanatory variables and an instrumental variable. We conduct a set of robustness checks to ensure the validity and reliability of our results.

Thirdly, this paper provides insights into potential strategies that China could adopt to reduce both emission intensity and total emission amount. Furthermore, the scheme and methodology in this study can be extended to more countries and extrapolated to a global scale. For countries with substantial regional disparities, our methodology can be utilized to investigate their regional carbon inequality. Analogous policies may then be contemplated to circumvent an imbalanced or polarized development pattern and reconcile the dilemma between economic development and carbon mitigation. From a global perspective, MNEs investing in technologically backward countries are strongly encouraged to transfer low-carbon technologies. This not only prevents international carbon leakages through FDI but also contributes to global carbon emission mitigation and inequality alleviation.

Therefore, we believe that the depth and breadth of this study is sufficient for publication in *Nature Communications*.

López, L.-A., Cadarso, M.-Á., Zafrilla, J. & Arce, G. (2019) The carbon footprint of the US multinationals' foreign affiliates. *Nature communications* 10, 1672.

Zhang, Z. et al. (2020) Embodied carbon emissions in the supply chains of multinational enterprises. *Nature Climate Change* 10, 1096-1101.

3.Changes in emission coefficients lead to changes in value added, which in turn leads to changes in carbon inequality. However, in the scenario setting, the authors did not consider the change in value added and considered it beyond the scope of the study. Therefore, the rationality and accuracy of the final carbon inequality cannot be guaranteed.

***Response:** Thank you for the comment. The design of the scenarios is based on a comprehensive consideration of practicality and feasibility. The rationale behind our scenario settings is as follows. We first conducted an econometric regression analysis, which demonstrated that the existence of MNEs significantly narrows the emission coefficient gaps between domestic-owned enterprises and MNEs. This indicates that the knowledge spillover effects from MNEs facilitate technological convergence, enabling lagging enterprises to improve their emission technology to catch up with that of industry leaders. For example, a company situated in an inland region could enhance its technology and reduce its emission coefficient by adopting advanced emission technology from multinational counterparts or similar companies in coastal region. Based on this solid empirical evidence, we designed four distinct scenarios, considering the potential upgrades in emission technology and adjusting the emission coefficients of technological laggards across different regions and industries.*

Meanwhile, we agree that the upgrading of emission technology may also increase value-added creation capacity. In such a case, emission technology improvements in inland regions could reduce carbon inequality to a larger extent than our scenario estimates indicate. Unfortunately, we lack quantitative evidence to determine the extent to which changes in emission coefficients lead to variations in value-added creation. Such changes could alter economic structures and reshape interregional input-output linkages. Including these channels arbitrarily in the simulation would contradict the fundamental premise of the IPIO-based model and introduce substantial uncertainty to the results. Therefore, we have consciously chosen not to include changes in the value-added coefficient or the input-output structure in our simulation and leave it for further exploration in future work.

In the revised manuscript, we have added a section discussing this limitation on page 27.

In summary, while the potential for carbon inequality reduction estimated by our scenario analyses may not be perfectly predicted, our scenario settings are grounded in the widely accepted theory of technology catch-up and supported by solid empirical evidence. Therefore, we believe that our scenario analyses yield rational and credible trends and implications.

4.This paper only considers the overall carbon inequality at the regional level, lacking an examination of the impact of multinational enterprises on the flow of carbon emissions and value added between regions.

Response: Thank you for the comment. Our calculations have already taken into account the impact of MNEs on the interregional flow of carbon emissions and value added. We employ an environmentally extended interprovincial input-output (IPIO) model which enables us to capture both direct and indirect interregional effects. In Figure 3, we present the contribution of MNEs to local value-added or carbon emissions, which gives the proportion of the overall contribution of MNEs to a province's value-added or carbon emissions relative to the province's value-added or carbon emissions. We also present the share of direct value-added or carbon emissions of MNEs in local value-added or carbon emissions, which gives the share of direct value-added or carbon emissions of MNEs within a province in the province's value-added or carbon emissions. Actually, the difference between these two indicators is the indirect impact of MNEs on a province via inter-regional input-output linkages.

Considering that the primary focus of this paper is carbon inequality, we believe it is unnecessary to provide extensive details on the inter-regional indirect impact of MNEs in the main text. Instead, we analyze the interregional carbon inequality across different regions which arises from the inter-regional impact of MNEs.

5. With the existing quality of this paper, I think it is more suitable for journals such as «Journal of Cleaner Production» and «Environmental Science & Technology», etc.

Response: We undertook a comprehensive literature review, including an examination of two relevant papers published in Nature journals. López et al. (2019) traced the carbon footprints of the US multinationals' foreign affiliates, while Zhang et al. (2020) traced the carbon footprints of MNEs on a global scale. As highlighted above, our study significantly expands and deepens the existing literature related to MNEs. Therefore, we believe it aligns well with the scope of Nature Communications.

López, L.-A., Cadarso, M.-Á., Zafrilla, J. & Arce, G. (2019) The carbon footprint of the US multinationals' foreign affiliates. *Nature communications* 10, 1672.

Zhang, Z. et al. (2020) Embodied carbon emissions in the supply chains of multinational enterprises. *Nature Climate Change* 10, 1096-1101.

Reviewer #2 (Remarks on code availability):

No code shared

Response: The relevant codes are shared on the Code Ocean platform.
<https://codeocean.com/capsule/5942607/tree>

Reviewer #3 (Remarks to the Author):

The authors studied the effects of MNEs on regional carbon inequality in China. Although I find the topic a trendy one (sustainability), I hope the comments below could help improve/reposition the paper.

Some statements are strong but unjustified. For example, "The presence of MNEs increased China's carbon inequality from 1997 to 2012." It could be a mere correlation as the HEM exercises do not guarantee a *ceteris paribus* interpretation.

Response: *Thank you for the comment. We have carefully checked and revised such unjustified statements throughout our manuscript. For example, we revised the sentence to "From 1997 to 2012, the presence of MNEs was a driving factor that exacerbated China's carbon inequality. Nevertheless, by 2017, a significant transformation occurred, with MNEs becoming a factor that mitigated this inequality" to avoid possible misunderstanding.*

Please refer to the updated manuscript for our revisions.

The instrumental variable (FMA) is a problematic one. Distance to the nearest port is far from random in this context, but is highly correlated with East/West or Coastal/Inland divide in China. A simple balance check could reveal this problem.

Response: *Thank you for the comment. The motivation of using an instrumental variable (IV) is to address potential endogeneity and reverse causality. For an IV to be effective, it must meet two primary conditions: relevance and exogeneity. In terms of relevance, the IV must be correlated with the endogenous explanatory variable, meaning that it should explain a significant portion of the variation in the endogenous variable. As for exogeneity, the IV must be uncorrelated with the error term in the original equation. In other words, the instrument should not directly affect the dependent variable except through the endogenous variable it seeks to instrument. Geographic variations are commonly employed as IV, meeting both relevance and exogeneity criteria.*

In our case, we choose the IV as province-level Foreign Market Access (FMA), which is measured by the distance from a province's capital city to its nearest port. Provinces closer to a port are more likely to have better access to foreign markets, thereby making them more appealing to MNEs. In other words, there is a correlation between FMA and the presence of MNEs (the explanatory variable). The IV is indeed highly correlated with East/West or Coastal/Inland divide in China, as it should be, since coastal provinces typically have more MNEs. Furthermore, a region's emission efficiency (the dependent variable) is not directly correlated with its geographical characteristics (distance to the nearest port). Therefore, we consider the FMA as an

appropriate IV. This IV (distance to nearest ports) was also employed in previous studies such as Redding and Venables (2004) and Storeygard (2016).

We also use lagged explanatory variables to address potential endogeneity and reverse causality and verify the robustness of the results by employing an alternative estimator-GMM to partially solve the endogeneity problem.

Redding, S., Venables, A.J. (2004) Economic geography and international inequality. *Journal of International Economics*, 62 (1), 53-82.

Storeygard, A. (2016) Farther on down the road: transport costs, trade and urban growth in sub-Saharan Africa. *The Review of Economic Studies*, 83 (3), 1263-1295.

What happened in (or around) 2017? 2017 has been highlighted throughout the paper as a year of pivotal change. But what exactly happened to the MNEs in China? Is this in line with the mechanism of carbon reduction? If so, any macro or micro level evidence?

Response: Thank you for the question. The effects of MNEs on China's carbon inequality can be ascribed to a combination of several important factors, including the geographical distribution of MNEs, the types of industries in which they operate, the differences in emission intensity between MNEs and domestic-owned enterprises, and regional disparities in emission intensity.

By 2017, the proportion of MNEs located in Western China had risen. Meanwhile, the share of MNEs in Western China operating in carbon-intensive industries had decreased to a lower level. These changes led to the overall emission intensities of MNEs in Western China becoming significantly lower than those of domestic-owned enterprises, thereby narrowing the emission intensity gap between Western and Eastern China.

These changes can be attributed to at least three factors. **First**, production costs—including labor, resources, and environmental costs—rose significantly in coastal provinces since the 2010s, prompting MNEs to relocate to inland regions where production costs are lower. **Second**, China implemented regional development strategies, such as 'The Great Western Development Strategy' and 'The Rise of Central China Strategy', to reduce regional disparities. Along with economic factors, these policies facilitated the relocation of MNEs to inland regions. **Third**, China formulated increasingly stringent and specific CO₂ mitigation policies after the 2010s. The '12th Five-Year Plan' set a target of a 17% reduction in CO₂ emissions per unit of GDP from 2011 to 2015. Additionally, the 'Plans for Greenhouse Gas Emissions Control in the 12th Five-Year Plan' issued in 2012 proposed some specific mitigation strategies, including stricter regulations on energy-intensive industries by raising entry thresholds. These tightening regulations accelerated the adoption of emission-reducing technologies by MNEs in China.

We have added these explanations on page 13. Please refer to the updated manuscript for our revisions.

The policy implications are limited (perhaps due to the descriptive results and the complex factors embedded into "MNEs"). The word "leveraging" in the title implies as if MNEs could be an exogenous force subject to policy manipulation. Unfortunately, the regional disparities in terms of natural and human resources are likely to be responsible for both carbon inequality and MNEs choices.

Response: *Thank you for the comment. It is true that the regional disparities in terms of natural and human resources are likely to be responsible for both carbon inequality and MNEs choices. In this paper, we do not stress that MNEs is an exogenous force subject to policy manipulation. Instead, we underscore the importance of sound policies in attracting and guiding MNEs decisions. For instance, cultivating a favorable business environment and augmenting incentives such as fiscal and taxation support can stimulate foreign investment.*

We use the word "leveraging" to describe utilizing the positive spillover effects of MNEs to reduce carbon emissions and inequality. Our quantitative analysis indicates that increasing MNE investment in inland regions can help lower carbon emissions and reduce regional inequality. However, achieving this transition requires supportive policies. That is why we highlight policy implications in the Discussion section.

*We provide policy implications from three key perspectives. **First**, from the national perspective, we discuss China's efforts not only to prevent the excessive exodus of MNEs but also to attract a growing influx of foreign investment. **Second**, from the regional perspective, we emphasize the need for intensified efforts to attract more foreign investment to inland China. **Third**, we underscore the imperative to enhance low-carbon technologies in inland regions, particularly within carbon-intensive industries. MNEs are suggested to share the carbon mitigation responsibility in their host regions and support the emission technology improvements of local enterprises.*

We also consider the possibility that MNEs may not increase their investment in inland regions as expected due to economic and geopolitical issues, thereby disrupting the technology spillover effects on local enterprises. Therefore, we analyze a scenario where domestic enterprises in inland regions converge to the emission technology levels of those in eastern regions. The results demonstrate a substantial decline in carbon emissions and an impressive reduction of in the C-Gini. We thus stress the importance of stimulating domestic-owned enterprises in coastal regions to increase their investments in clean technologies inland if foreign investments would be disrupted.

Further proofreading:

For example, the authors state, on page 11, that "However, the economic growth rate in the west did not match the pace of emission growth". However, the mismatch was observed in the east (e.g., Jiangsu with high economic growth but moderate emission growth).

Response: Thank you for the comment. We have carefully done thorough proofreading and revised any improper wording. For example, we revised the mentioned sentences to “However, the ratio of economic growth to emissions growth in the west was lower than that in the east. For instance, from 1997 to 2017, Xinjiang’s value-added increased by 928.6%, while its CO2 emissions rose by 617.7%. In contrast, Jiangsu experienced a substantial 1,132.0% increase in value-added over the same period, accompanied by a comparatively moderate rise of 305.5% in CO2 emissions.”.

Response to the Reviewers

From: “Leveraging multinational enterprises to reduce the escalating regional carbon inequality in China” (NCOMMS-24-21432B)

The authors are grateful for the insightful comments and constructive suggestions from the reviewers. We have carefully revised the manuscript accordingly, which we believe has made for a better article. Please find below our detailed point-by-point responses to the reviewers. To enhance readability, we have copied the feedbacks of the reviewers and put our responses in italics.

REVIEWER COMMENTS

Reviewer #1 (Remarks to the Author):

The authors have further revised to increase the readability of the figures and include more information on the methods and limitations. The author has responded well to all my comments. I would also like to thank the authors for their efforts.

Response: Thank you very much for helping improve the paper.

Reviewer #2 (Remarks to the Author):

At this revised manuscript authors developed a substantially improved study, comparing to the initial study, having the review comments addressed in a systematic and responsible manner. My recommendation is therefore for the paper to be accepted for publication.

Response: Thank you very much for helping improve the paper.

Reviewer #3 (Remarks to the Author):

While the authors' efforts to revise the paper are appreciated, several concerns remain.

1. Central Issue

The argument that MNEs are a driving factor behind carbon inequality is unconvincing. MNEs might not be an appropriate focal variable for deriving meaningful policy implications.

***Response:** Thank you for the comment. This paper aims at estimating how and to what extent MNEs in China contribute to its regional economic development (provincial value-added creation), carbon emissions, and the ensuing regional carbon inequality over the past decades. As is widely recognized, a primary motivation for countries to attract MNEs is to foster economic growth. That is why we firstly measure the contribution of MNEs to China's value-added and find that MNEs contributed a considerable proportion (from 19.7% in 1997 to 26.2% in 2017) of its national value-added (shown in Fig. 1). Meanwhile, MNEs are responsible for a significant proportion (20-30%) of China's carbon emissions. Furthermore, we find that MNEs exert substantial yet unequal economic and environmental effects across different provinces.*

*Therefore, we zoom in analyzing the effects of MNEs on carbon inequality by defining the inequality as the generated emissions across the provinces relative to their value-added gains. Our findings reveal that **MNEs exerted nonnegligible effects on carbon inequality**, altering the C-Ginis by 10%-20% during 1997-2017. We have conducted a DM (Diebold-Mariano) test and confirmed that these effects were statistically significant.*

*We acknowledge that multiple driving factors may influence carbon inequality. However, based on the findings of this study, we emphasize that **MNEs represent a nonnegligible yet frequently overlooked factor in shaping carbon inequality**. Their role, as demonstrated in our analysis, is both substantial and deserving of greater attention in discussions of regional economic and environmental disparities.*

*Our research on MNEs provides multiple meaningful policy implications. **First**, we find that the increasing presence of MNEs in inland regions has contributed to balanced economic development and mitigated regional carbon inequality. We thus emphasize the need for intensified efforts to attract more foreign investment to Central, Western, and Northeastern China. Inland provinces are strongly suggested to improve their governance practices to cultivate a favorable business environment and augment incentives such as fiscal and taxation support to attract foreign investors. **Second**, our scenario analyses underscore the imperative to enhance low-carbon technologies in inland regions, particularly within carbon-intensive industries. MNEs should share environmental responsibility in their host regions and strengthening coordination between MNEs and local enterprises is imperative to expedite the diffusion of cleaner production technology. **Third**, our accounting work suggests that MNEs generate a significant proportion of CO₂ emissions in China, and therefore, they should share the*

carbon mitigation responsibility. We emphasize that establishing an accurate carbon emission accounting system such as the one we proposed in this paper is a crucial step forward in assigning the mitigation responsibilities to each region, key industries and enterprises, thereby supporting China's dual efforts in controlling emission amount and intensity.

2. Econometric Concerns

There are several econometric issues that need to be addressed:

Many important covariates are not controlled for, such as urbanization, administrative quality, and other potential factors.

The IV "FMA" is likely correlated with these omitted covariates, meaning it may influence the dependent variable (emissions) through channels other than "MNEs". This violates the "exogeneity" condition.

Consider replacing "MNEs" with an unrelated variable, such as "number of foreign tourists". The IV test might still be passed, and significant results might still be obtained. However, it would be unreasonable to suggest mitigating carbon inequality by increasing the number of foreign tourists.

Response: Thank you for the comment. We included a series of control variables in the regressions. Considering that the main channels through which a region's emission performance is determined are scale effects, sectoral composition effects, and technique effects, we introduce control variables including production scale, industrial structure, and technology innovation level. We also introduced urbanization level and environmental regulation level, the latter of which also serves as a proxy for administrative quality. These controls ensure that our analysis comprehensively addresses the multifaceted factors influencing regional emission performance. We include a set of industry fixed effects, province fixed effects and time fixed effects to account for possible country, industry and time unobserved heterogeneity, and thus, we argue that omitted variables bias is not a (big) concern.

The motivation of using an instrumental variable (IV) is to address potential endogeneity and reverse causality. For an IV to be valid and effective, it must satisfy two key conditions: relevance and exogeneity. In terms of relevance, the IV must be correlated with the endogenous explanatory variable, meaning that it should explain a significant portion of the variation in the endogenous variable. As for exogeneity, the IV must be uncorrelated with the error term in the original equation. In other words, the instrument should not directly affect the dependent variable except through the endogenous variable it seeks to instrument.

Geographic variables are frequently employed as IVs because they often satisfy both relevance and exogeneity criteria. In this study, we utilize a geographic variable—Foreign Market Access (FMA), the distance from a province's capital city to its nearest port—as the IV. Provinces closer to a port are more likely to have better access to foreign markets, thereby making them more appealing to MNEs. In other words, there

is a correlation between FMA and the presence of MNEs (the explanatory variable). Furthermore, a region's emission efficiency (the dependent variable) is not directly correlated with its geographical characteristics. Therefore, *this variable is well-suited for the IV role because geographic characteristics are unlikely to be influenced by other economic or social factors, and thus is very unlikely correlated with other potential (if any) omitted variables.* In contrast, variables like the “number of foreign tourists” are more susceptible to influence by other factors, making them unsuitable as IVs. This explains why geographic variables are a popular and reliable choice for instrumental variables in empirical research.

Similar studies that have employed geographic variables as instrumental variables (IVs) include several influential papers published in top-tier economic journals. For instance, Redding and Venables (2004) utilized geographic variables to examine the role of market access in determining economic geography and trade patterns. Similarly, Storeygard (2016) leveraged geographic instruments to study the impact of transportation infrastructure on economic development in sub-Saharan Africa. These studies demonstrate the effectiveness of geographic variables in addressing endogeneity concerns. Their successful application in high-impact research underscores the robustness and reliability of geographic instruments in empirical economic analysis.

We also employ another frequently used IV—the lagged explanatory variable—to address potential endogeneity and verify the robustness of the results. Lagged variables are frequently employed as IVs because they are often correlated with the current values of the explanatory variable (satisfying the relevance condition) while being less likely to be influenced by contemporaneous shocks or reverse causality (satisfying the exogeneity condition). The dual approach—combining geographic variables and lagged explanatory variables—strengthens the robustness of our results.

We provided a more detailed introduction to the regression model, along with detailed results for the control variables, in the Supplementary Information.

Redding, S., Venables, A.J. (2004) *Economic geography and international inequality.* *Journal of International Economics*, 62 (1), 53-82.

Storeygard, A. (2016) *Farther on down the road: transport costs, trade and urban growth in sub-Saharan Africa.* *The Review of Economic Studies*, 83 (3), 1263-1295.

3. Policy Implications

If MNEs are not an exogenous force, the validity of policy recommendations based on MNEs is questionable. Mitigating emission inequality might be more effectively achieved by promoting low-carbon technologies and enforcing stringent CO₂ mitigation policies for domestic enterprises (mentioned by the authors), without involving MNEs.

Response: Thank you for the comment. We agree that strategies such as promoting low-carbon technologies and enforcing stringent CO₂ mitigation policies for domestic enterprises may effectively mitigate emission inequality. These approaches have already been highlighted in our discussion. Nevertheless, in this paper we also emphasize the potential of MNEs in mitigating emission amount and inequality as our findings reveal that MNEs are instrumental in advancing low-carbon technologies.

In economic and environmental studies, MNEs are often treated as exogenous forces when they are perceived as external factors influencing a system from outside its boundaries. In this study, we focus on the impact of MNEs on domestic regional economies and environments. MNEs can be considered exogenous in this context because their operations and strategic decisions are determined by their foreign headquarters. Nevertheless, it should be noted that these decisions are not made in isolation; they can be influenced by a range of external and internal factors. For instance, bilateral or multilateral political relationships, as well as domestic considerations such as production costs and the local business environment, may shape the strategies and actions of MNEs. Thus, while MNEs may initially appear as exogenous forces, their integration into and responsiveness to local and global dynamics complicate this characterization.

In formulating our policy implications, we have carefully accounted for these complexities. By acknowledging the dual influence of global strategies and local conditions on MNE behavior, our recommendations are proposed to be both reasonable and well-suited to addressing the multifaceted role of MNEs in shaping regional economic and environmental outcomes.

4. The Role of 2017

It remains unclear why the year 2017 is considered pivotal:

Was the observed increase in the proportion of MNEs located in Western China due to more MNEs relocating to this region, or was it the result of coastal MNEs moving abroad?

Development plans such as the Western Development and the Rise of Central China began in the 2000s. Why would their effects only materialize around 2017?

The authors also mentioned the spillover effects of MNEs, but there seems to be no direct quantitative evidence supporting this mechanism. Also, if spillover effects are indeed a key mechanism, similar effects should be observable within domestic enterprises, at least in recent years.

Response: Thank you for the comment. The effects of MNEs on China's carbon inequality can be ascribed to a combination of multiple factors. One key factor is the shifting proportion of MNEs located in inland regions. The observed rise in the share of MNEs in Western China can be attributed to two concurrent trends: an increase in MNEs relocating to this region and coastal MNEs moving their operations abroad. Of

these, the former is the more dominant driver, as evidenced by the overall increase in the MNEs entering China by 2017 (Chen et al. 2023). This suggests that the relocation of MNEs to inland areas, rather than their exit from the country, has played a more significant role in reshaping regional carbon inequality. These dynamics highlight the importance of understanding how MNE distribution patterns influence carbon emissions and regional disparities in China.

We provide three factors influencing the distribution of MNEs: production costs, environmental policies, and development plans. Among these, production costs are the most dominant driver, as they directly affect the operational decisions of MNEs. Environmental policies also play a significant role, particularly as China has implemented increasingly stringent regulations to curb emissions, prompting MNEs to adapt their strategies. In contrast, development plans such as the Western Development Strategy primarily target domestic enterprises and, as a result, have a much more limited impact on MNEs. Since the 2010s, production costs in coastal provinces have risen significantly, and stricter environmental regulations have been implemented, particularly in these regions. While development plans have been in place since the 2000s, their influence has been relatively limited compared to the other two factors. As a result, the proportion of MNEs in inland regions has gradually increased since the 2010s. However, due to the availability of novel interprovincial input-output (IPIO) tables only for 2012 and 2017 since the 2010s, the effects of these changes are primarily reflected in the results for 2017.

We provide quantitative evidence for the spillover effects of MNEs by conducting a regression analysis where the explanatory variable is the presence of MNEs and the explained variable is the ratio of the emission coefficient of domestic-owned enterprises to that of MNEs within the same industry (the gap of emission coefficient between domestic-owned enterprises and MNEs). The results show that MNEs significantly reduce the gaps, supporting the mechanism. We agree that spillover effects also exist among domestic enterprises. Therefore, we stress the importance of stimulating domestic-owned enterprises with advanced technologies in coastal regions to increase their investments in clean technologies in inland areas. Such initiatives would facilitate the faster transfer of knowledge and advanced emission-reducing technologies to these regions.

Please refer to the updated manuscript for our revisions.

Response to the Reviewers

From: “Leveraging multinational enterprises to reduce the escalating regional carbon inequality in China” (NCOMMS-24-21432C)

The authors are grateful for the insightful comments and constructive suggestions from the reviewers. We have carefully revised the manuscript accordingly, which we believe has made for a better article. Please find below our detailed point-by-point responses to the reviewers. To enhance readability, we have copied the feedbacks of the reviewers and put our responses in italics.

REVIEWER COMMENTS

Basically, as far as I am concerned, questions 1-3 from referee#3 seem to be well addressed (or at least have tried their best), while question 4 could actually be improved with additional structural analysis.

1. Reviewer 3’s major concern is that MNEs is not the focal driving factor behind carbon inequality. As I understand, the authors are not arguing that MNEs are the sole factor and they have already replied that “We acknowledge that multiple driving factors may influence carbon inequality. However, based on the findings of this study, we emphasize that MNEs represent a nonnegligible yet frequently overlooked factor in shaping carbon inequality”. Thus, they are trying to show that the MNEs are one of the key factors which have been overlooked in previous analysis, and that MNEs have a significant effect on the carbon inequalities.

Response: Thank you for your thoughtful feedback. You’ve perfectly captured the essence of what we intended to convey.

2. As far as I understand, the reviewer’s major concern is that the identified effect on MNEs using the IV could be a correlation instead of causality. For instance, the coefficient might also be significant if the question is “does the number of foreign tourists—instead of MNEs—influence carbon inequality”, which is actually unreasonable to suggest mitigating carbon inequality by increasing the number of foreign tourists. In the response letter, the authors are trying to show that the IVs they selected, either Foreign Market Access or the lagged explanatory variable, are reasonable IV choices. These IVs have been used in previous studies, like JIE (2004) and RES (2016). Actually, it is always difficult to find a “perfect” IV, Actually, it is always difficult to find a “perfect” IV, as most IVs rely on theoretical assumptions that cannot be fully tested empirically. However, what matters is whether the chosen IVs

are reasonable, widely accepted in the literature, and supported by robustness checks. The authors have made efforts to justify the validity of their IVs by drawing on prior influential studies and explaining why their instruments satisfy the relevance and exogeneity conditions.

Response: *Thank you for your thoughtful feedback. You've perfectly captured the essence of what we intended to convey.*

3. Policy Implications

This question seems to be already well addressed.

Response: *Thank you very much.*

4. The role of 2017

As for this issue, I do agree with the reviewer's point that the role of 2017 and the different role of MNEs in previous year and in 2017 is not very easy to explain or understand. To better identify the role of MNEs across all these years clearly, they would better use methods like Structural Decomposition Analysis to quantitatively reveal the structural changes and the major factors behind them.

Response: *Thank you for the constructive suggestion. We have accordingly employed a Structural Decomposition Analysis (SDA) to investigate the driving factors behind the changes in MNEs' contribution to the C-Gini between time periods.*

The contribution of MNEs to the C-Gini coefficient (\hat{G}) is calculated as the difference between the original and a hypothetical Gini coefficient: $\hat{G} = \tilde{G} - G$. The original C-Gini (G) is derived from the original provincial carbon emissions and value-added, while the hypothetical C-Gini (\tilde{G}) is calculated using the hypothetical values of provincial carbon emissions and value-added which are provided by equation (5).

The value of \hat{G} depends on the direct carbon emission coefficient \mathbf{e} , the value-added coefficient \mathbf{v} , the input matrix \mathbf{A} , and the final demand \mathbf{f} . The matrix \mathbf{A} is split into a part that gives the overall production technology (\mathbf{A}^), reflecting the total direct inputs that are required per unit of output, and the other part that gives the regional shares ($\mathbf{\Phi}$), which captures the structure of inter-regional intermediate inputs. The matrix \mathbf{A}^* is defined as*

$$\mathbf{A}^* = \begin{bmatrix} \mathbf{A}_{*1}^{DD} & \mathbf{A}_{*1}^{DH} & \mathbf{A}_{*1}^{DF} & \cdots & \mathbf{A}_{*r}^{DD} & \mathbf{A}_{*r}^{DH} & \mathbf{A}_{*r}^{DF} \\ \mathbf{A}_{*1}^{HD} & \mathbf{A}_{*1}^{HH} & \mathbf{A}_{*1}^{HF} & \cdots & \mathbf{A}_{*r}^{HD} & \mathbf{A}_{*r}^{HH} & \mathbf{A}_{*r}^{HF} \\ \mathbf{A}_{*1}^{FD} & \mathbf{A}_{*1}^{FH} & \mathbf{A}_{*1}^{FF} & \cdots & \mathbf{A}_{*r}^{FD} & \mathbf{A}_{*r}^{FH} & \mathbf{A}_{*r}^{FF} \\ \vdots & \vdots & \vdots & \vdots & \vdots & \vdots & \vdots \\ \mathbf{A}_{*1}^{DD} & \mathbf{A}_{*1}^{DH} & \mathbf{A}_{*1}^{DF} & \cdots & \mathbf{A}_{*r}^{DD} & \mathbf{A}_{*r}^{DH} & \mathbf{A}_{*r}^{DF} \\ \mathbf{A}_{*1}^{HD} & \mathbf{A}_{*1}^{HH} & \mathbf{A}_{*1}^{HF} & \cdots & \mathbf{A}_{*r}^{HD} & \mathbf{A}_{*r}^{HH} & \mathbf{A}_{*r}^{HF} \\ \mathbf{A}_{*1}^{FD} & \mathbf{A}_{*1}^{FH} & \mathbf{A}_{*1}^{FF} & \cdots & \mathbf{A}_{*r}^{FD} & \mathbf{A}_{*r}^{FH} & \mathbf{A}_{*r}^{FF} \end{bmatrix},$$

with the sub-matrix $\begin{bmatrix} \mathbf{A}_{*r}^{DD} & \mathbf{A}_{*r}^{DH} & \mathbf{A}_{*r}^{DF} \\ \mathbf{A}_{*r}^{HD} & \mathbf{A}_{*r}^{HH} & \mathbf{A}_{*r}^{HF} \\ \mathbf{A}_{*r}^{FD} & \mathbf{A}_{*r}^{FH} & \mathbf{A}_{*r}^{FF} \end{bmatrix} = \begin{bmatrix} \sum_{s=1}^{31} \mathbf{A}_{sr}^{DD} & \sum_{s=1}^{31} \mathbf{A}_{sr}^{DH} & \sum_{s=1}^{31} \mathbf{A}_{sr}^{DF} \\ \sum_{s=1}^{31} \mathbf{A}_{sr}^{HD} & \sum_{s=1}^{31} \mathbf{A}_{sr}^{HH} & \sum_{s=1}^{31} \mathbf{A}_{sr}^{HF} \\ \sum_{s=1}^{31} \mathbf{A}_{sr}^{FD} & \sum_{s=1}^{31} \mathbf{A}_{sr}^{FH} & \sum_{s=1}^{31} \mathbf{A}_{sr}^{FF} \end{bmatrix}$. These

matrices give the aggregated amounts of intermediate inputs per unit of output irrespective of their geographical sources. The matrix Φ is given by $\Phi = \mathbf{A} // \mathbf{A}^*$, where $//$ denotes element-wise division. Similarly, the final demand (\mathbf{f}) can be split into the total final demands (\mathbf{f}^*) and the regional structure (Π) of final demands.

The value of \hat{G} can then be written as a function of six matrices and vectors: $\hat{G} = f(\mathbf{e}, \mathbf{v}, \mathbf{A}^*, \Phi, \mathbf{f}^*, \Pi)$. The changes in \hat{G} can be decomposed into the contributions of six driving factors using the average of two polar decompositions:

$$\Delta \hat{G} = \Delta \mathbf{e} + \Delta \mathbf{v} + \Delta \mathbf{A}^* + \Delta \Phi + \Delta \mathbf{f}^* + \Delta \Pi,$$

where $\Delta \mathbf{e}$, $\Delta \mathbf{v}$, $\Delta \mathbf{A}^*$, $\Delta \Phi$, $\Delta \mathbf{f}^*$, and $\Delta \Pi$ give the contribution of the direct carbon emission coefficient, the value-added coefficient, the overall production technology, the structure of inter-regional intermediate inputs, the total final demands, and the regional structure of final demands, respectively. The changes in the geographical distribution of MNEs are reflected in both $\Delta \Phi$ and $\Delta \Pi$, while $\Delta \mathbf{A}^*$ and $\Delta \mathbf{f}^*$ also capture the changes in the structures of intermediate products and final products provided by MNEs, respectively.

Our results show that MNEs reduced C-Gini by 2017 while amplified it during 1997–2012. Using the SDA, we identify that the main driving factors behind this shift are changes in both the geographical distribution of MNEs and the structure of their products. In 1997, only 2.2% of MNEs located in Western China (6.1% in Central China and 7.0% in Northeastern China), contrasting sharply with the dominant presence of 84.7% of MNEs concentrated in Eastern China. Moreover, in Western China, a larger proportion of MNEs (32.6%) operated within carbon-intensive industries compared to domestic-owned enterprises (23.5%). As a result, MNEs in Western China had higher

overall emission intensities than domestic-owned enterprises. In contrast, MNEs in Eastern China had significantly lower overall emission intensities compared to domestic-owned enterprises. This widened the emission intensity gap between Western and Eastern China. This pattern persisted from 1997 to 2012, contributing to exacerbating carbon inequality during this period.

By 2017, the proportion of MNEs located in Western China had risen to 9.3%. Meanwhile, the share of MNEs in Western China operating in carbon-intensive industries had decreased to a lower level (16.2%). These changes led to the overall emission intensities of MNEs in Western China becoming significantly lower than those of domestic-owned enterprises, thereby narrowing the emission intensity gap between Western and Eastern China. Consequently, the presence of MNEs became a factor that alleviated China's carbon inequality by 2017.

The significant changes in MNEs' geographical distribution and product structures observed since the 2010s are the combined consequences of both economic factors and policy interventions. First and foremost, production costs rose significantly in coastal regions since the 2010s, prompting MNEs to relocate to inland regions where operational expenses remained lower. Second, China's increasingly stringent climate policies have incentivized cleaner production. For example, the '12th Five-Year Plan' set a target of a 17% reduction in CO₂ emissions per unit of GDP from 2011 to 2015. Additionally, the 2012 'Plans for Greenhouse Gas Emissions Control in the 12th Five-Year Plan' proposed some specific mitigation strategies, including stricter regulations on energy-intensive industries by raising entry thresholds. These tightening regulations accelerated the adoption of emission-reducing technologies among MNEs and encouraged them to supply cleaner products. Third, China implemented regional development strategies, which also facilitate the relocation of MNEs to inland regions.

We have added the SDA method and the explanations. Please refer to the new manuscript for our revision.